behaviour/evolution/cognition

macaque, evolution of cognition, sociality, reversal-learning task, distraction task, go/no-go task

**Author for correspondence:**
Louise Loyant
e-mail: louise.loyant@port.ac.uk

# Heterogeneity of performances in several inhibitory control tasks: male rhesus macaques are more easily distracted than females

Louise Loyant[1], Bridget M. Waller[2], Jérôme Micheletta[1] and Marine Joly[1]

[1]Centre for Comparative and Evolutionary Psychology, Department of Psychology, University of Portsmouth, King Henry Building, Portsmouth PO1 2DY, UK
[2]Department of Psychology, Nottingham Trent University, Nottingham, UK

LL, 0000-0003-0816-2736; BMW, 0000-0001-6303-7458; JM, 0000-0002-4480-6781; MJ, 0000-0002-0784-5167

Inhibitory control, the ability to override a dominant response, is crucial in many aspects of everyday life. In animal studies, striking individual variations are often largely ignored and their causes rarely considered. Hence, our aims were to systematically investigate individual variability in inhibitory control, to replicate the most common causes of individual variation (age, sex and rank) and to determine if these factors had a consistent effect on three main components of inhibitory control (inhibition of a distraction, inhibition of an action, inhibition of a cognitive set). We tested 21 rhesus macaques (*Macaca mulatta*) in a battery of validated touchscreen tasks. We first found individual variations in all inhibitory control performances. We then demonstrated that males had poorer performances to inhibit a distraction and that middle-aged individuals exhibited poorer performance in the inhibition of a cognitive set. Hence, the factors of age and sex were not consistently associated with the main components of inhibitory control, suggesting a multi-faceted structure. The rank of the subjects did not influence any inhibitory control performances. This study adopts a novel approach for animal behaviour studies and gives new insight into the individual variability of inhibitory control which is crucial to understand its evolutionary underpinnings.

# 1. Introduction

To live in a complex social environment, animals need to manage their impulsive behaviours to maintain group cohesion and survival. For example, when a higher ranked social partner is present, an individual might benefit withholding inappropriate behaviours, such as behaving aggressively, when competing over food [1,2] or a mating partner [3,4]. Such cognitive processes are defined as inhibitory control, the ability to override an impulsive, automatic or pre-learned response in order to maximize fitness benefits [5–7]. Inhibitory process allows to flexibly respond to the environment and adjust behaviours becoming counterproductive or potentially harmful [8]. Inhibitory control has been associated with positive life outcomes in humans [9–12], and brain size [13–15] or problem-solving skills in animals [16–18]. Such inhibitory process, crucial in any individual's day-to-day life, is thus essential during complex social interaction and problem solving [16–19]. Here, we focused on the individual variation in the major components of inhibitory control: distraction inhibition (e.g. control of a distractor to focus on a goal), action inhibition (e.g. inhibition of a dominant motoric action) and cognitive set inhibition (e.g. inhibition of a pre-learned behaviour; [5,7,19,20]).

While in human psychometric research, individual differences in cognition have been studied since the early twentieth century [21–24], research on non-human animals' cognition has, until recently, tended to minimize the importance of variation among individuals [23,24]. Performances of individuals are often pooled together implying that the performance of a sample of a population are representative of the whole species [23–27]. Besides, studies often focus on a subset of high-performing individuals to draw conclusions on the presence or absence of a cognitive capacity in a species [24,26]. In inhibitory control research, large comparative studies have demonstrated high variability in inhibitory control between species [14,28]; however, striking individual variations are usually ignored and considered as mere 'noise' around the population mean [23,26]. Nonetheless, over the last decade, researchers have started to focus on intra- and inter-individual variations in cognitive abilities and the factors generating individual differences (for review see [23,24,26,27]). Increasingly, authors are recognizing that valuable information can be lost when focusing only on group-level performances [23]. Several studies put a great effort in casting light on individual differences in animal cognition [24–26,29]; however, only a handful of them have systematically examined individual differences in inhibitory control, and the findings are contradictory. For instance, in guppies (*Poecilia reticulata*), researchers reported individual differences in two measures of inhibition of action. In these tasks, subjects needed to inhibit reaching directly for a prey through a transparent glass tube [30–32] or through a transparent cylinder [32]. Similarly, pheasants (*Phasianus colchicus*) displayed a great individual variability in a response inhibition task in which the subjects were required to adjust their movement while pursuing a moving target [33]. Dogs also demonstrated individual differences in common inhibitory control tasks such as the detour task [34]. However, another study in pheasants did not demonstrate a stable individual variation using common detour-reaching tasks [35]. Similarly, in a meta-analysis, Völter *et al*. [27] re-analysed, from an individual difference perspective, two large comparative studies [14,28]. These studies measured the inhibition of action (a detour-reaching task) and the inhibition of a cognitive set (the A-not-B task in which the subjects are required to inhibit a previously rewarded behaviour to learn a new reward contingency) in several species. From the first study [14], the researchers extracted and re-analysed performances of 15 species, and from the second study, they re-analysed performances of six primate species [28]. The authors did not find consistent individual differences between these tasks of inhibitory control. To the best of our knowledge, individual differences in the inhibition of distraction in non-human animals have not yet been specifically studied. These contradictory findings could indicate a need for a more systematic analysis of individual differences across all the main components of this crucial ability.

Even though some authors put great efforts in studying individual difference in inhibitory control, the causes of these variations are seldom investigated and remain largely unknown [26]. Yet, recent studies have started to look at the factors influencing inhibitory control variability [26,30,35–38]. At the time scale of an individual's life, individual variation in inhibitory performances can be influenced by several factors, such as the characteristics of an individual (e.g. age, sex or rank). Inhibitory control develops from early life to adulthood through neurogenesis and experience [7], and demonstrates a high plasticity [39,40]. Cognitive abilities are predicted to improve with experience over an individual's lifetime (for a review see [41]). For instance, researchers found that in chimpanzees, age had a positive effect on individual performance in a physical cognitive task (a causality task and a repeated spatial memory task) [42]. Similarly, older chimpanzees and bonobos performed better in

physical tasks such as locating a reward with changing location [43]. Thus experience, acquired over an individual's life, can dramatically influence subjects' cognitive performances, such as inhibitory control [44]. For instance, ravens demonstrated a gradual increase in performance in the cylinder task through age [45]. Similarly, dogs which already experienced an opaque cylinder were more successful at the transparent cylinder task [46]. Previous experience with transparent surfaces improved pheasant's inhibitory control on a novel detour task [47]. It was also demonstrated that hand tracking experience affects A-not-B task success in New Caledonian crows [48]. Similarly, brown lemurs used previous knowledge to master a novel reverse-reward task through generalization [49]. It was also demonstrated that dogs used social learning to improve their detour task performances [50]. More generally, it was suggested that previous life's experiences explained why pet dogs outperformed shelter dogs on an A-not-B task [51]. It is thus possible that successful inhibitory performances are facilitated by an individual's life experience and could improve with age until the end of adulthood.

Variation in inhibitory control is also influenced by the sex of the subject. Human studies demonstrated that women outperform men on go/no-go tasks (in this task subjects need to respond to frequently presented stimuli while withholding prepotent response to infrequently presented non-target) [52,53]. In animals, male guppies had worse performances than females in an inhibition of action task, the transparent tube task, in which males attempted to attack the prey inside a transparent tube twice as often as females [30]. Female guppies were also better at reversing a pre-learned rule [37]. This sex difference is particularly strong in one measurement of distraction inhibition, the emotional Stroop task (an emotional pictorial stimulus interferes with the goal of a task; [54]), with men having an attentional bias toward threatening stimuli [55,56]. In animal research, results are not revealing clear sex differences. For example, Boggiani *et al.* [57] found no sex difference in the distractive effect of a threatening stimulus in capuchins monkeys (*Sapajus apella*). Besides, several studies found a strong bias toward threatening stimuli in male rhesus macaques (*Macaca mulatta*, [58–60]), unfortunately these results were not compared with female performances.

Finally, the structure of the social environment where an individual interacts is also an important factor influencing variations in cognitive abilities [61,62]. The social intelligence hypothesis [61,62] postulates that the demands associated with a complex social life, with differentiated social relationships, generate selection for increased brain size and higher cognitive performances, such as inhibitory control (e.g. [63]). In a more complex environment, individuals may tune their social behaviours in relation to the identity of the social partner with whom they interact [1,2,63–65]. Such skills are particularly adaptive in societies where dominance hierarchies determine access to food and mates [1,2,66]. Research on non-human primates suggests that living in social systems, with steeper dominance hierarchies, may be associated with better inhibitory control skills [1,65]. A subordinate individual may frequently inhibit his impulsive behaviours (reaching for food or a mating partner) in the presence of more dominant individuals to avoid conflicts [1,2,28,65]. Johnson-Ulrich & Holekamp [36] have demonstrated that lower ranked hyenas (*Crocuta Crocuta*) living in larger groups have better inhibitory control than higher ranked conspecifics in a common task of inhibition of action, the cylinder task (subject needs to inhibit reaching directly for food through the transparent surface of a cylinder). To the best of our knowledge, this rank effect on inhibitory control has only been studied in one study using a cylinder task [36] but this social factor could potentially affect all the components of inhibitory control. To summarize, depending on the task, individual's characteristics (age, sex and rank) can have a strong influence on variations of inhibitory control skills, but this hasn't been examined systematically.

Besides, it is still unclear if individual factors such as sex and rank are consistently influencing the different inhibitory processes. For example, if the factor age influences one measure of inhibition of action it should equally influence one measure of inhibition of a cognitive set if inhibitory control is one common ability. However, the findings are mixed. In dogs, Bray *et al.* [67] found that the age influenced dogs' performances in the cylinder task but not in the A-not-B task. Contrarily, in the same species, Vernouillet *et al.* [46] found an effect of age on performances in a modified version of the A-not-B task but not in the original A-not-B task nor in the cylinder task. The same study found no effect of the sex for the A-not-B task, but they found a higher score for females in the cylinder task. Hence, the influence of different factors on the three main components of this ability remains unclear.

Therefore, the aim of this study was threefold: (i) to systematically demonstrate individual variability in the three main components of inhibitory control in non-human primates, (ii) to investigate the most common causes (age, sex and rank) of these individual variations, and (iii) to determine if these influencing factors have coherent effects on the three main components of inhibitory control.

Rhesus macaques (*Macaca mulatta*) represent an interesting model to investigate the factors influencing inhibitory control. This species, phylogenetically close to human species, live in large complex social groups [68]. These groups are characterized by multi-male–multi-female organization with steep hierarchy organized in matrilines [69]. Rhesus macaques present sex differences in social behaviour, anatomy and biology [70,71]. They can live, in the wild, up to 25 years old [68]. Besides, macaque species possess enhanced general intelligence compared with other mammals and can perform the cognitively demanding tasks [72].

We used a battery of three reliable and valid touchscreen tasks [73]. To investigate *inhibition of distraction*, we conducted a distraction task (experiment 1). In this task, a subject must inhibit a dominant and prepotent response to a distractor [58,74–78]. To investigate *inhibition of action*, we conducted a go/no-go task (experiment 2). Here, a subject learns to develop a prepotent motor response to frequently appearing target and must withhold it to less frequently appearing non-target [5,19,20,68,74,75]. Lastly, to assess *inhibition of a cognitive set*, we conducted a reversal-learning task (experiment 3). In this task, a subject must inhibit a pre-learned rule to adopt a new set of rules [48,67,74,75].

We expected individual variability in the inhibitory control performances for the three tasks. We hypothesized that the different factors would all have an effect on the three main components of inhibitory control. More specifically, we predicted that the females would have better inhibitory control performances than males. We expected that as subjects grow older, they will get more experience and will have better inhibitory performances. Finally, we predicted that lower ranked individuals would have better inhibitory control than higher ranked conspecifics.

# 2. General methods

## 2.1. Subjects

All the 21 subjects (12 males, 9 females; aged from 3 to 17 years old, $M = 8.1$, s.d. $= 4.05$) participating in this study were adult rhesus macaques (*Macaca mulatta*) housed in the breeding colony of the Medical Research Council Centre for Macaques (MRC-CFM) in Porton Down, United Kingdom. The subjects were taken from 12 mixed groups constituted of 9–20 individuals with a mean of 15 individuals. The groups were constituted of one dominant male and several females and younglings. One group was constituted of only males and one group of only females. Each group had access to a free-roaming room with a large bay window allowing a natural day–night cycle and an adjacent cage area. Rooms were enriched with climbing devices. The floor was covered with a deep layer of straw and shavings. All rooms were temperature controlled (20°C ± 5) with humidity at 55% ±10. Subjects received a supply of dried forage mix (cereal, peas, beans, etc.), a range of fruit and vegetables, bread and boiled eggs. Water was available ad libitum. All subjects had access to food and water prior to and during the experiment. Hierarchy, calculated in each group using David's scores [79], was provided by the head of research of the facility (see electronic supplementary materials 1 and 2; [81]). Agonistic behaviours including threats, displacements, chases and physical conflict were recorded to assess the hierarchy. The caretakers regularly monitored the groups, and David's scores were updated accordingly. Using video recordings of training and test sessions, a blind observer coded agonistic interactions between the tested individual and other conspecifics to verify the given ranks for each group (see electronic supplementary material 1, table S1; [81]). Eighteen of the subjects had already participated in one behavioural study involving looking at pictures [80]. However, none of them had experience with cognitive or touchscreen experiments. Thirty-one macaques started the touchscreen training phases (see electronic supplementary material 1; [81]) but only 21 (12 males and 9 females) successfully completed the training phases and were able to take part in the experiment. Among the tested individuals, 5 females and 10 males were high-ranking.

## 2.2. Apparatus

To minimize stress, all tests were conducted on a voluntary basis, in the enclosure, with no isolation from the social group. Outside the cage, a laptop was connected via HDMI and USB cables to a capacitive touchscreen (ELO 1590 L, 19″ in diagonal, resolution 1280 × 1024 pixels, frequency of 60 Hz) (figure 1). The program Elo touch solution 6.9.20 was used for calibration. All experimental procedures including stimulus presentation and response collection (response latency in ms: time to complete a trial and success, i.e. touching the correct target within the time limit) were carried out using Matlab coding

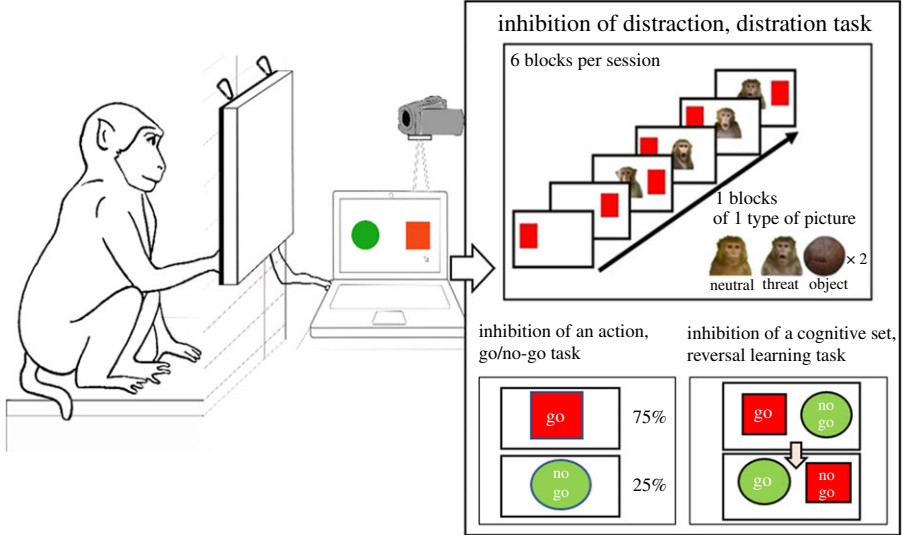

**Figure 1.** Schematic representation of the touchscreen apparatus and inhibitory control tasks procedure. The distraction task (inhibition of distraction, a testing block of six trials is presented, a session is composed of six blocks with three different types of picture as distractors), the go/no-go task (inhibition of action) and the reversal-learning task (inhibition of a cognitive set) are presented.

program (v. R 2018b, using Psychtoolbox-3.0.15 functions, see [81] for Matlab scripts). The Matlab scripts were specifically conceived for the needs of this study; an individual progression file allowed the experimenter to abort and come back to the same point of a running session. If a trial was aborted, the response latency recorded was deleted. The computer gave auditory feedback in response to the subject's performance. All sessions were videotaped with one digital video camera (Sony HDR-CX330EB). The rewards (small pieces of apple, dry raisins and peanuts) for each correct answer were given by hand.

## 2.3. General procedure

The experimenter launched the task and entered the name of the subject. When more than one individual per cage were tested and when the rest of the group was interacting with the apparatus, a research assistant was distracting the other macaques at the other side of the room while the subject was tested. Every session was initiated by the subject touching a red cross located in the centre of the screen, starting the experiment and the time recording. The session was aborted when another individual displaced the tested individual or interacted with the touchscreen. There was a 5–10 min break between each session to allow the subjects to refocus on the task. If the subject left the testing area or was not focusing attention on the screen, the session was aborted. If the target was not touched within the specified time limit (see task descriptions below for specific time limit), the timer was paused, the response latency was not recorded and a red cross appeared in the centre of the screen until the session was resumed by touching it. If the subject stayed inactive for more than 5 min or lost its attention, the experiment was stopped and the remaining sessions were finished the next testing day; if the subject did not participate for three testing days in a row the subject was excluded from the task. The tasks were conducted in the same order as they were built upon the previous task: first the distraction task (task only requiring touching a target as in the training phases), second the go/no-go task (a novel unrewarded stimulus was introduced) and finally the reversal-learning task (built upon the previously rewarded and unrewarded stimuli). Once all the sessions of a task were completed the next task was conducted on the next testing day.

## 2.4. Statistical analysis

To study the relationship between the outcome variables of inhibitory control (see each task for specific outcomes) and the different categories of explanatory variables, we used linear mixed models (LMM, for repeated continuous outcomes) or general linear mixed models (GLMM, for binomial outcomes). For the

individual characteristics, models were fitted with the following variables: sex (male or female), age (in years) and rank (low or high). We hypothesized that as subjects get older, they get more experience in inhibitory control, so we used the age of the subject as a proxy for their experience. The oldest monkey in this study was 17 years old, which can be considered middle age. We conducted the rank analysis only on females as most of the groups were constituted of several matrilines with only one male. This configuration allowed us to have access to females with more diverse ranks but did not allow the investigation of rank effects for males. The last explanatory variable category was experimental factors to control for habituation and learning: trial number, session number and the type of stimulus depending on the experiment. The random factor of the individual identity remained in all models to account for repeated measures of individuals (see [81] for the dataset generated during the study and for R scripts used in the analysis).

We used LMM and GLMM using the functions 'lmer' and 'glmer' from the R package 'lme4' v. 1.1-21 [82]. For binomial models' convergence, we used bound optimization by quadratic approximation (BOBYQA) with a set of 100 000 iterations [83]. The models were fitted using the maximum-likelihood (ML) function. We used the functions 'hist' and 'qqnorm' (from the package 'stats' v. 3.6.2) to visually check for the normal distribution of the residuals. For binomial or Poisson distribution, we used the function 'simulateResiduals' (from the package 'DHARMa', [84]). Models were compared by the likelihood ratio test using the function 'anova' from the R package 'car' v. 3.0-6. Significant effects on the models were considered if the model with the predictor was significantly different from the model without it.

We applied backward reduction to analyse the contribution of each variable on the models [85]. Initially, all explanatory variables and interaction were fitted in the maximal model. Non-significant interaction and terms on the model ($p > 0.05$) were dropped sequentially in $p$-value decreasing order to simplify the model. Once an optimum model was obtained with only variables having a significant effect on the model, we compared the effect of each variable by comparing the optimum model and the model without this variable. We presented the model with all the predictors. Our significant threshold was $p < 0.05$.

To study individual differences, we used the repeatability estimates (R) which provide a way of assessing individual differences by quantifying the amount of variation explained by inter-individual variation of performances in the tasks relative to intra-individual variation (developed by [86]; e.g. used in [30,80,87,88]). We used the function 'rpt', from the 'rptR' package v. 0.9.22 in R [86,89]. We applied a restricted ML function and the individual identity was specified as a random intercept effect. We estimated 95% CI with 1000 bootstraps and 1000 permutations. The appropriate type of data distribution was adjusted in each model regarding the dependent variable investigated ('Gaussian', 'Poisson' or 'Binomial'). This function showed whether the individual macaques' performances were significantly repeatable across trials and sessions as expected for individual differences. To maximize individual variation within individuals, we only included subjects which performed the tasks on different days ($N = 16$ for the distraction task, $N = 15$ for the go/no-go task and $N = 19$ for the reversal-learning task). An individual's performance was considered as repeatable and significant if the $p$-value from the likelihood ratio test was less than 0.05. We then adjusted the repeatability models with the significant factors obtained from the LMM and GLMM analysis to obtain the *adjusted repeatability*, $R_{\mathrm{adj}}$. The adjusted repeatability is an estimate that adjusts for confounding effects by removing fixed effect variance from the estimate [88,89].

# 3. Experiment 1: inhibition of a distraction, the distraction task

The emotional Stroop task is used in human research in which subjects are required to name the colours of words that differ in emotional valence, with a longer response latency to negative words [90,91]. Instead of emotional words we used pictorial stimuli presented at the same time as a target (as in [77] in chimpanzees and in [59] with rhesus macaques) and looked at the distraction effect (i.e. the effect on the response latency) of each kind of stimulus (an object, a neutral social stimulus and a threatening social stimulus) on the performances on a goal-oriented task. We hypothesized that, in this distraction task, the picture of a threatening conspecific would particularly decrease the performance of the subjects (i.e. longer response latency) more than the control trials (trials without pictures), the pictures of a neutral conspecific or an object as previously shown in the same species [59,92]. We also hypothesized that this attentional bias would be stronger in males compared with

females [54–56]. We finally hypothesized that older individuals and lower ranked individuals would be less distracted by the pictures as they are supposed to have better inhibitory control skills [36,42,43,45].

## 3.1. Subjects

The 21 subjects who progressed from the touchscreen training phases (see electronic supplementary material 1, [81]) participated in the distraction task. This resulted in a sample of 12 males and 9 females (age ranging from 3 to 17 years old).

## 3.2. Stimuli

The subject had to touch a target on the screen (a red rectangle of $10 \times 13$ cm) while distractors consisted of three different stimulus categories ('Object', 'Neutral' conspecific and 'Threatening' conspecific) appeared in the centre of the screen. The 'Control' trials only consisted of the target presented, without any picture displayed. All the pictures of $16 \times 18.5$ cm (matched for colour, contrast and the luminosity using the function 'Match colour' in Adobe Photoshop). The category 'Object', similar to objects already present in the subject's environment, included a leather ball, a leather bag, a brown stone and a wooden log (see [81] for stimuli). The conspecific pictures were chosen to be as realistic as possible, in actual size, depicting a frontal view of the face and the torso of four unknown adult rhesus macaque (one picture of a male and three of females). The 'Neutral' conspecific included four pictures of individuals with a neutral facial expression (see [81] for stimuli). The 'Threatening' conspecific included four pictures showing a 'open mouth threat' facial expression (see [81] for stimuli), frequently displayed by rhesus macaques [58,59,93]. Several studies have demonstrated that rhesus macaques perceive, at first, pictorial stimuli of macaques as real conspecifics. For example, naive rhesus monkeys typically react to pictures of conspecifics with retreat, threat responses and vocalizations ([94,95], for review see [96]).

## 3.3. Design

Before starting the experiment, subjects were required to pass a refresher test with 80% of success (10 trials with only the target displayed). Every session was initiated by the subject touching a red cross in the centre of the screen. Once the subject touched the target, a high-pitched chime (composed of three sound frequencies: 800, 1300 and 2000 Hz) was played, the timer was stopped and the reward was given. After an inter-trial of 2000 ms, with only the white background displayed, the next trial was presented. The target was randomly displayed at the far left and right of the screen. The distractor appeared, at the same time, at the centre of the screen. Each session of 36 trials consisted of six blocks. Each block, divided in six trials, started with two trials with the target but without any pictures ('Control' trials). Following this, a block of four pictures of the same category was displayed (figure 1; each picture was seen in total six times over the three sessions). Each block and trial were counterbalanced across subjects. In previous studies, it appeared that with this block presentation, the Stroop effect was more pronounced. This block presentation has been adopted in humans [76,90,97] and in chimpanzees [77]. Based on a pilot study on the distraction task ($N = 4$ subjects), we set up a maximum response time of 35 s. This time period allowed the subject to display a behavioural response, control it and continue the task. If the subjects did not touch the target within this time limit, its response time was not included in the analysis.

## 3.4. Analysis

For the distraction task, we computed a *distraction control score*, representing the difference between the mean response latency in 'Control' trials for each individual minus the response latency in one trial with pictures. A higher score would indicate better control of a distraction, as the subject's reaction to the stimuli interfered less with the goal of the task. As the distribution of the residuals for the distraction control score was not satisfyingly following a normal distribution, we applied the following transformation: (normalized distraction control score $= \sqrt{(\max(\text{distraction control score} + 1))} -$ distraction control score) as advised in [85,98] for moderately negatively skewed distribution.

We included order of the blocks within a session as an explanatory variable to control for habituation between blocks (order of blocks from 1 to 6). We looked at the effect of the type of picture on the distraction control score (picture of an object, picture of a neutral conspecific face, picture of a

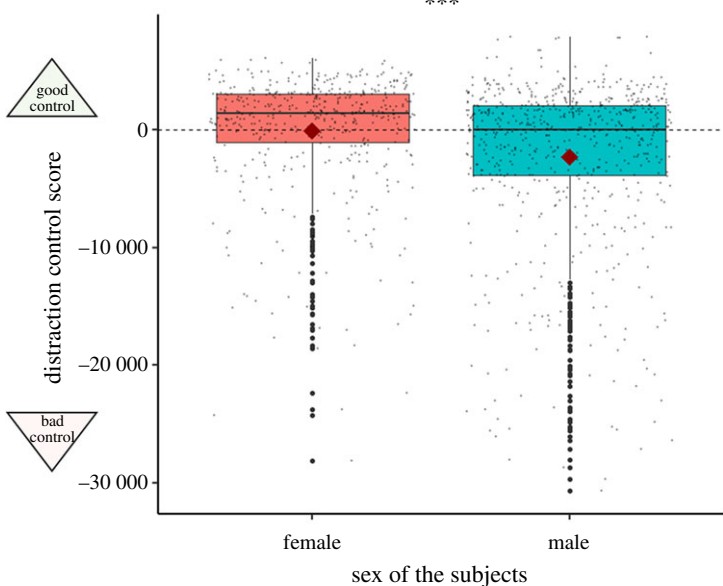

**Figure 2.** Distraction control score between females and males when all types of pictures were pooled together in the distraction task. Males had a lower distraction control score than females, as they were more distracted by pictures. Diamonds represents the mean. Horizontal lines represent the 25th, 50th and 75th percentile, and the whiskers extend to 1.5 inter-quartile range. ***$p < 0.001$.

threatening conspecific face). We also looked at the interaction between sex and type of stimulus (either a picture or no picture) and between sex and the type of picture (object versus face and threatening versus neutral face) as the sex of the subject has been shown to interact with his reaction to a stimulus [54–56]. Thirteen subjects already took part in a behavioural study involving looking at a picture in 2018 [80], but the effect of this experience did not significantly influence the model, so we did not include this factor into the further analysis. Type of picture was nested within the individual identity. We used the *post hoc* test Tukey's honest significant difference test, to analyse the difference between each type of pictures [99], using the function 'glht' from the R package 'multcomp' v. 1.4-13 with type of picture as an explanatory factor of the model. We also recorded the number of facial expressions of the subject toward a specific stimulus.

## 3.5. Results

Individual macaques exhibited significant repeatability of their distraction control score across trials and session ($R = 0.114$, CI $= [0.041, 0.199]$, $p < 0.001$). This result demonstrates an individual variation in their distraction task performances (see visual variation in performances in the figure 2). When adjusting for the following confounding factors session, trial, order of the blocks and interaction between sex and type of stimulus, the macaques' performances were still repeatable ($R_{adj} = 0.119$, CI $= [0.04, 0.001]$, $p < 0.0001$). This result demonstrates an individual variation in the distraction task performances even when taking into account confounding variables.

When all the sessions were taken altogether, there was a significant main effect of the interaction between the sex and the presence or absence of a picture, ($\chi^2_3 = 51.517$, $N = 21$, $p < 0.001$, see electronic supplementary material 1, table S2, [81]), when pictures were presented (all types of pictures taken together), females had a higher distraction control score ($M = -74.64$ ms, s.d. $= 5162.78$, $N = 9$) than males ($M = -2302.66$ ms, s.d. $= 6935.3$, $N = 12$, figure 2). This result means that females were less distracted by the pictorial stimuli, in general, than males (table 1). However, the specific type of the picture (either a picture of a face versus a picture of an object or a picture of a threatening face versus a picture of a neutral face) did not have an effect on the distraction control score (see table 1).

There were significant main effects of the trial ($\chi^2_1 = 21.408$, $N = 21$, $p < 0.001$), the number of the block ($\chi^2_1 = 21.802$, $N = 21$, $p < 0.001$) and the session ($\chi^2_1 = 10.931$, $N = 21$, $p < 0.001$) on the performances (see electronic supplementary material 1, table S2; [81]). Macaques demonstrated better control scores as trial and sessions increased (table 1). The other explanatory variables did not have a significant effect on the models (see electronic supplementary material 1, table S2; [81]). We conducted the rank analysis only for

**Table 1.** Results of the LMM for the normalized distraction control score (distraction task) when all the sessions and sex were taken together. Explanatory variables were divided into three categories: individual characteristics (sex and age), experimental determinants (type of picture, trial, number of the block and session) and interaction sex and type of stimulus (control versus a picture). All full models included the type of picture nested in the individual ID as a random factor. The estimates (representing the change in the dependent variable relative to the baseline category of each predictor variable), standard error, $t$-value and $p$-value using ML method. The variables presence of a picture, trial number, order of blocks, session and the interaction between sex and stimulus had a significant effect on the models. $^{***}p < 0.001$, $^{**}p < 0.01$, $^{*}p < 0.05$.

| distraction control score | all sessions | | | |
|---|---|---|---|---|
| predictor | estimate | s.e. | $t$-value | Pr(>|t|) |
| (intercept) | 944.548 | 673.015 | 1.403 | 0.169 |
| sex (male) | 342.457 | 619.875 | 0.552 | 0.585 |
| picture versus no picture | −1119.518 | 510.116 | −2.195 | 0.034* |
| picture of an object versus a face | 17.212 | 131.233 | 0.131 | 0.896 |
| picture of a threat versus a neutral face | −3.794 | 230.481 | −0.016 | 0.987 |
| age | −48.645 | 60.957 | −0.798 | 0.435 |
| trial | 367.160 | 72.446 | 5.068 | $4.418 \times 10^{-7}$*** |
| number of the block | −2147.928 | 426.758 | −5.033 | $5.291 \times 10^{-7}$*** |
| session | 527.563 | 143.731 | 3.670 | $2.488 \times 10^{-4}$*** |
| sex (male) × stimulus (picture) | −2420.388 | 624.197 | −3.878 | 0.001** |

females (see electronic supplementary material 1, table S3-a; [81]). For the females, there was also a significant main effect of the trial ($\chi^2_1 = 7.773$, $N = 9$, $p < 0.05$), the number of the block ($\chi^2_1 = 8.89$, $N = 9$, $p < 0.05$) and the session ($\chi^2_1 = 9.768$, $N = 9$, $p < 0.05$). But the other variables including the rank ($\chi^2_1 = 0.024$, $N = 9$, $p = 0.87$) did not have a significant effect.

Besides, as the block number within a session also influenced the distraction control score, we look at the performances when the pictures were seen the first time (for the block 1 as in [77]). The control score for the 'Threatening' condition ($M = -4112.85$ ms, s.d. $= 8955.85$) was significantly lower than the other conditions ('Neutral' condition $M = -714$ ms, s.d. $= 6042.5$; 'Object' condition $M = 173.15$ ms, s.d. $= 5248.85$; 'Control' condition $M = 145.64$, s.d. $= 4660.44$; see electronic supplementary material 1, S4-a; [81]; figure 3).

As the interaction between the sex of the subject and the type of stimulus had an effect on the performances, we simplified the analysis by looking at each sex separately (table 2). For males only, all sessions altogether, the type of picture had a significant main effect on the distraction score ($\chi^2_1 = 26.553$, $N = 21$, $p < 0.001$, see electronic supplementary material 1, table S3-b; [81]). When we looked precisely of the effect of each type of picture, using the Tukey HSD test indicated that the mean distraction control score for the 'Control' condition ($M = -0.002$, s.d. $= 4636.38$) was significantly higher than the 'Threatening' condition ($M = -2086.42$ ms, s.d. $= 7347.66$), the 'Neutral' condition ($M = -2440.35$, s.d. $= 6859.59$) and the 'Object' condition ($M = -2365.96$ ms, s.d. $= 6640.65$, see electronic supplementary material 1, table S4-b; [81]). For the females, all sessions together, none of the variables had a significant effect on the models (table 2; electronic supplementary material 1, table S4-c; [81]).

Regarding the number of facial expressions produced by the subjects in response to conspecific stimuli, six macaques (six males) expressed a submissive facial expression 'bared teeth' toward the threatening stimulus, four macaques to the neutral stimulus (three males, one female) and none towards the objects or the control stimulus.

## 3.6. Discussion

As expected, we found a significant individual variation in the inhibition of distraction. We also found, as hypothesized, that males were more distracted by pictorial stimuli than females. For the first exposure to pictures, males and females were more distracted by the threatening stimulus. We did not find any effect of the age or the rank (in females) on the inhibition of a distraction.

We first found individual differences in the performances of inhibition of distraction, even when considering confounding factors. To the best of our knowledge, this is the first time individual differences

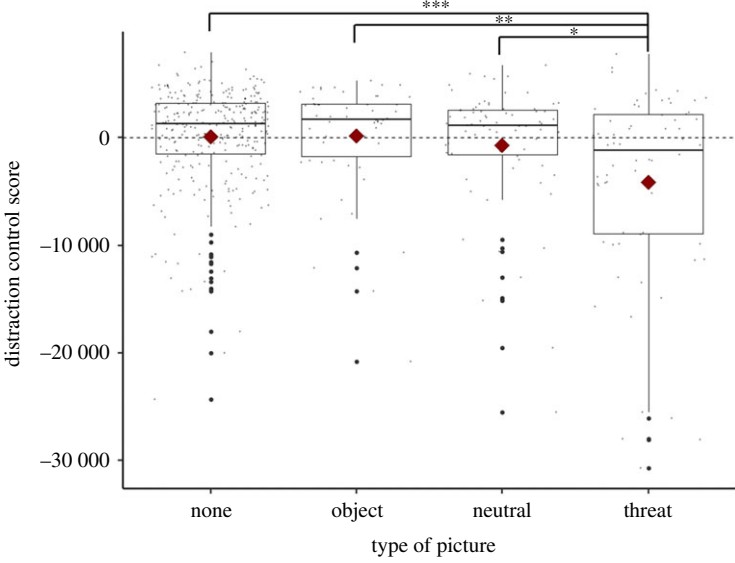

**Figure 3.** Distraction control score in the distraction task for males and females (all sessions, block 1), when looking at each type of picture (control: no picture, object: picture of an object, 'Object' condition; neutral: picture of a conspecific neutral face, 'Neutral' condition; and threat: picture of a conspecific threatening faced, 'Threatening' condition). Subjects had a significantly lower distraction score in the 'Threatening' condition compared with any other type of condition ('Control', 'Neutral' or 'Object' condition). Diamonds represent the mean. Horizontal lines represent the 25th, 50th and 75th percentile, and the whiskers extend to 1.5 inter-quartile range. $^{***}p < 0.001$, $^{*}p < 0.05$.

**Table 2.** Results of the LMM for the distraction control score (normalized) for males and females all sessions together for the distraction task. Explanatory variables were individual characteristics (age and rank), the type of stimulus (control versus a picture), trial, number of the block and session. All full models included the type of picture nested in the individual ID as a random factor. The estimates (representing the change in the dependent variable relative to the baseline category of each predictor variable), standard error, $t$-value and $p$-value using ML method. The variables trial, order of blocks, session and presence of a picture (in males) had a significant effect on the models. $^{***}p < 0.001$, $^{**}p < 0.01$.

| distraction control score | females | | | | males | | | |
|---|---|---|---|---|---|---|---|---|
| | estim. | s.e. | $t$-value | Pr(>\|t\|) | estim. | s.e. | $t$-value | Pr(>\|t\|) |
| intercept | 89.087 | 3.517 | 25.331 | 0.000 | 89.184 | 4.383 | 20.346 | 0.000 |
| pictures versus no picture | 3.789 | 2.606 | 1.454 | 0.146 | 15.067 | 2.4547 | 6.138 | 0.000*** |
| age | −0.149 | 0.379 | −0.394 | 0.702 | 0.607 | 0.358 | 1.696 | 0.117 |
| rank (high) | 0.442 | 2.484 | 0.178 | 0.863 | | | | |
| trial | −1.841 | 0.512 | −3.593 | 0.000*** | −1.476 | 0.456 | −3.238 | 0.001** |
| number of the block | 11.303 | 3.022 | 3.741 | 0.000*** | 8.439 | 2.682 | 3.146 | 0.002** |
| session | −3.585 | 0.953 | −3.760 | 0.000*** | −1.153 | 0.944 | −1.222 | 0.222 |

in distraction task have been systematically analysed in non-human animals. It would be interesting to re-analyse, from an individual difference perspective (as did Völter *et al.* [27] for the cylinder task and the A-not-B tasks), datasets from other studies on the inhibition of distraction (e.g. [58,59,77]).

Macaques demonstrated better control scores as trials and sessions increased through habituation. Similar to the current study, in [94,95], rhesus macaques first reacted with vocalizations, threats and retreats to the first presentation of slides of conspecifics' faces. However, this behaviour did not persist, as macaques realized that the stimuli were only pictures. Fagot *et al.* [96] described in their model, a first phase of confusion between the picture and reality, but after repeated exposure, the animal could then consider the picture as equivalent to the represented object or process the picture and the object as independent.

As expected, we found that males had lower performances in the distraction inhibition compared with females. When all the sessions were taken together, males were more distracted by the pictures. In the literature, better inhibitory control skills in females were also found in the go/no-go task [52,53] and in other tasks of inhibition of an action such as the cylinder task [32] and the tube task [31,32]. Similar findings were also found in a task of cognition inhibition, the reversal-learning task [37]. Concerning the distraction task, this sex difference was only found in human research, with male being more distracted than female [55,56]. To the best of our knowledge, this study is the first one demonstrating a sex difference in performances in the distraction task in non-human primates. Hence, males and females may face different selective pressures on inhibitory control, which can lead to different fitness consequences [30,100,101]. For instance, female guppies had better performances at the reversal-learning task [37,101], at the cylinder task and at the tube task [30]. The authors proposed that this sex difference could be explained by a selection in males for high persistence and reduced inhibition in the mating behaviour. Regarding the Stroop task, this attentional bias toward pictorial stimulus could be explained by a higher level of vigilance needed from males at the top of the hierarchy. For instance, Watson *et al.* [102] found that rhesus macaque's vigilance while drinking from a waterhole was higher for males.

This sex difference in inhibitory control could be explained by difference in hormones levels. For instance, in human research, testosterone level is associated in males with a greater attention to negative social cues [103,104]. In animals, male rhesus macaques' testosterone level significantly increased watching time of video clips which depicted fights between unfamiliar conspecifics [105]. However, the effect of testosterone on attentional bias toward negative stimuli was not found with pictorial stimuli [106]. In another study, testosterone level was associated with a greater impulsivity in male rats [107]. Hence, higher testosterone levels could explain a worse inhibitory control performances in our male subjects.

For the first appearances of the pictures, when both sexes were taken together, the macaques were more distracted by the threatening stimulus than other types of distractor (objects and neutral conspecific faces), showing that the emotional content of the picture is important. This attentional bias toward threatening stimuli was similarly found in chimpanzees, using pictures of a veterinarian [77] and in rhesus macaques, using pictures of a threatening conspecific [58,59]. This prioritization of attention to threat has been proposed as an evolutionary adaptation [108]. Our results could then be explained as a survival mechanism as it would be more adaptive to focus more on a threat as it appears. Besides, macaques, and particularly males, were making fear-related or submissive facial expressions toward conspecific faces and mainly toward threatening faces. Similarly, emotional response to threatening stimulus on a screen was also found in chimpanzees with an increasing heart rate when pictures of aggressive conspecifics appeared [109], and with changes in skin temperature in viewing emotionally negative videos [110]. This emotional reaction could demonstrate that some of our subjects confused the pictures with real individuals, processing them as a threat. Our results seem to indicate that some rhesus macaques we tested are, at first, not understanding the concept of a picture [96], possibly as a result of limited exposure to this type of stimuli.

Neither the age of the subjects nor their rank had an effect on their distraction inhibition. In the literature, to the best of our knowledge, none of these factors have been considered in primate studies using pictures as distractors [58,59,77]. The absence of effect could either be due to a true absence of influence of these factors on the inhibition of a distraction or it could also be due to our small sample size, reducing the power of our statistical analysis.

# 4. Experiment 2: inhibition of action, the go/no-go task

In the go/no-go task, the subjects need to respond to frequently presented stimuli while withholding prepotent response to infrequently presented no-go stimuli [5,7]. We predicted that older monkeys, with improved inhibitory control skills, will be better at controlling their impulsive actions (as in [42–45]). Besides, males will be more impulsive than females, as they were described as less capable of controlling a dominant response [52,53]. Finally, low-ranked individuals, which must frequently inhibit both feeding and aggression in the presence of higher ranking conspecifics were predicted to have better inhibitory control skills [36].

## 4.1. Subjects

The same subjects who completed the distraction task participated in the go/no-go task. However, one male subject was not willing to continue the experiment. This yielded to a sample of 11 males and 9 females (age ranging from 3 to 17 years old, mean age in years $M \pm$ s.d. $= 8.65 \pm 4.31$, $N = 20$).

**Table 3.** Results of the LMM and GLMM for the go/no-go task: (*a*) for the probability of success, or (*b*) for the log of response latency on a no-go trial. Explanatory variables were individual characteristics (sex and age), trial and session. All full models included the individual ID as a random factor. The estimates (representing the change in the dependent variable relative to the baseline category of each predictor variable), standard error, *t*-value, *z*-value and *p*-value using ML method. None of the explanatory variables had a significant effect on the models.

| no-go trial | (*a*) success | | | | (*b*) log(response latency) | | | |
|---|---|---|---|---|---|---|---|---|
| predictor | estimate | s.e. | *z*-value | Pr(>|z|) | estimate | s.e. | *t*-value | Pr(>|t|) |
| (intercept) | −0.226 | 0.456 | −0.496 | 0.620 | 7.874 | 0.101 | 77.849 | 0.000 |
| sex (female) | 0.017 | 0.390 | 0.044 | 0.965 | 0.036 | 0.077 | 0.465 | 0.648 |
| age | −0.001 | 0.044 | −0.019 | 0.985 | 0.008 | 0.009 | 0.937 | 0.361 |
| trial | 0.004 | 0.006 | 0.635 | 0.525 | −0.001 | 0.001 | −0.547 | 0.585 |
| session | 0.034 | 0.048 | 0.712 | 0.476 | 0.008 | 0.010 | 0.843 | 0.399 |

## 4.2. Stimuli

The 'go' stimulus was a red rectangle of 16 × 18 cm. The 'no go' stimulus was a green (RBG 150, 255, 150) circle of 16 × 16 cm.

## 4.3. Design

The apparatus and the general procedure were identical to the distraction task. A red rectangle (go stimulus) and a green circle (no-go stimulus) appeared randomly (with no more than two 'no-go' in a row) at the middle of the screen, with 75% of 'go' stimuli during the session to elicit a prepotent response toward the screen. The 'go' stimulus was preceded by a 600 ms high-pitched sound and the 'no-go' by a 600 ms low-pitched sound. The 'go' stimulus stayed on the screen until it was touched. We set up a maximum response time (i.e. touching the 'go' stimuli) of 15 s; after this the red cross appeared on the screen and the response variables were not recorded. The 'no-go' stimulus disappeared if not touched during 2000 ms and the subject received a reward. If the 'no-go' stimulus was touched during this lapse of time, an 'incorrect' sound (with frequency 800, 1300, 2000 Hz) was produced, and the reward was not given, a blank white background appeared for 3000 ms. If the screen was touched outside the stimulus no sound was produced and the trial continued. At first, we fixed a success criterion for the subject's performances at 80% of correct trials per session, but four macaques never reached this criterion. The performance was therefore measured on a fixed number of sessions (i.e. five sessions of 40 trials).

## 4.4. Analysis

For the second task, the go/no-go task, to quantify the individual's ability to inhibit its prepotent action, we recorded the *success* and the *response latency* in a trial when a no-go was present. We analysed the responses for the five sessions. A higher probability of success and a longer response latency for no-go trial would indicate an individual better at inhibiting the action. Response latency variables were log transformed to approach a normal distribution of the residuals [111].

## 4.5. Result

Individual macaques exhibited significant repeatability in the go/no-go accuracy performance ($R = 0.089$, CI = [0.024, 0.172], $p < 0.001$) and in the response latency ($R = 0.104$, CI = [0.042, 0.187], $p < 0.001$) on a no-go trial, across trials and session. These results mean that there are individual variations in the accuracy and in the response latency in the go/no-go task. None of the individual characteristics (age, sex and rank) or the task determinants (trial and session) had a significant main effect on the probability of success or on the log of the response latency on a no-go trial, so the R estimates were not adjusted (table 3; electronic supplementary material 1, tables S5 and S6; [81]). Thus, the inhibition of action of our subjects was not influenced by any of the factors we controlled for. The rank in females did not have a significant main effect on the probability of success ($\chi^2_1 = 0.055$, $N = 9$, $p = 0.81$), nor on the log

of the response latency on a no-go trial ($\chi^2_1 = 0.057$, $N = 9$, $p = 0.81$, see electronic supplementary material 1, tables S7, S8 and S9; [81]). However, the probability of success did increase with the number of the session ($\chi^2_1 = 4.659$, $p = 0.03$, $N = 9$; see electronic supplementary material 1, table S7 and S8). The females only had a better accuracy on a trial as the number of the sessions increase (see electronic supplementary material 1, tables S7, S8 and S9; [81]). This improvement of performances over sessions was not found when only males were considered (see electronic supplementary material 1, tables S7, S8 and S9 [81]).

## 4.6. Discussion

As expected, we found a significant individual variation in the inhibition of an action. However, none of the studied factors had an effect on the subjects' performances (males and females pulled together). However, only females improved their accuracy over sessions.

We first found that macaques exhibit strong individual variations in inhibiting an impulsive action in both their response time and their accuracy to realize the task. For instance, for the last session, three macaques were above 80% of success (i.e. not touching the no-go stimulus significantly above chance) and five below 50% of success. If the task was too easy for all the individuals, variation in individual accuracy could not be detected. These results demonstrate the importance to design a task difficult enough to reveal difference between individuals (a signature limit of performances, [27]).

When males and females were pooled together, we demonstrated that the go/no-go task was not sensitive to any individual group or experiment determinants, neither when considering the macaques' accuracy nor their response latency. In contrast with our results, human studies demonstrated that women outperform men on the no-go trials, indicating greater inhibition [52,53]. However, human studies typically benefit from a larger sample size (40 women and 39 men in [53]; 15 women and 15 men in [52]), which can increase the likelihood of finding a significant effect [27,112]. However, other human studies found no sex difference in the performances in the go/no-go task, but interestingly, they found differences in the pattern of regional brain activation [113]. Maybe in our study the sex difference in rhesus macaques could be found if we increase our sample size or if we look at difference in brain activation (with a PET-scan for example). Surprisingly, a study in guppies also demonstrated sex differences in tasks of inhibition of action, the cylinder and the tube task [30], with a sample of only 14 females and 14 males. Authors suggested that fish might not possess the same cognitive process or neural circuits underlying inhibition as mammals. This sex difference in guppies might be due to a specific mating system, the males having been selected for reduced behavioural inhibition [30].

However, when females were considered alone, there was an improvement of their performances over sessions. Contrarily to females, males did not improve their performances at all. Hence, the sex difference could be found in the way subjects learn to respond to the task, with females learning how to inhibit their action more efficiently over time.

Contrary to our results, some studies found that older and more experienced primates were better at several cognitive tasks [42,43]. Unfortunately, to the best of our knowledge, the factor age has not been analysed for the go/no-go task in the animal literature. An explanation for our absence of age difference could be that the simple variant of the task we chose was too difficult for the more experienced individual to show cognitive improvement. The small sample size in our study compared with sample size in human research can also be an explanatory factor for this lack of age effect.

Unfortunately, to the best of our knowledge, the effects of the rank were not investigated on the go/no-go task. It was only investigated in spotted hyenas in the cylinder task [36]. We would potentially need to increase the sample size to reveal this rank effect, or it might also be specific to the cylinder task.

# 5. Experiment 3: inhibition of a cognitive set, the reversal-learning task

In the reversal learning, the subjects first learn a stimulus-reward contingency (acquisition rule). Once a pre-specified criterion is reached, this first association is reversed (reversed rule). Subjects must then inhibit a prepotent response to previously correct stimuli and shift responses to a new stimulus-reward contingency. We expected that females will be better at this reversal-learning task as in [37]. Older subjects will be better at reversing a rule as they have better cognitive capabilities [42,43] and will make less perseveration errors (i.e. the repetition of a response toward the wrong stimulus even if the reward-stimulus contingency has changed).

## 5.1. Subjects

The same 20 subjects (11 males and 9 females) which completed the go/no-go task participated in the reversal-learning task. One individual never learnt the acquisition rule.

## 5.2. Stimuli

At first the 'correct' stimulus was the same red rectangle as before. The 'incorrect' stimulus was a green circle of 15 x 15 cm.

## 5.3. Design

The apparatus and the general procedure were identical to the go/no-go task. However, at the beginning, two stimuli were displayed at the same time on the screen: a 'go' stimulus (the red rectangle) and a 'no-go' stimulus (a green circle). The side on which the stimuli were presented was counterbalanced across trials. When the subject touched the 'go' stimulus, the usual 'correct' sound was played, a reward was given and a new trial began. If the subject touched the incorrect stimulus, an incorrect sound was played, and the two stimuli stayed on the screen until the correct stimulus was touched. If the background was touched nothing happened. A session consisted of 40 trials. We set up a maximum response time of 15 s. When the macaque performed correctly 75% of 20 trials (touched the correct stimulus from the first attempt; the criterion was lowered from 80 to 75% because of an error in the Matlab script), the rule was reversed: the correct stimulus became the incorrect and the incorrect the correct. One male macaque did not reach the first criterion and was excluded from the study. The reversed session was continued until the success criterion was reached again (75% of success for the whole session). The number of taps on the wrong stimulus was recorded as well as the response latency on correct trials.

## 5.4. Analysis

For the reversal-learning task, individuals were required to inhibit a response that was previously successful. As a measurement of inhibitory control, we recorded the accuracy on a trial for the first (acquisition rule) and second rule (reverse rule) and the number of trials to learn the rules. We also recorded the perseveration error (number of taps when the wrong stimulus is displayed) and the response latency on a correct trial. To approach a normal distribution of the residuals, the response latency was log transformed [111] and the number of taps were log10 transformed [98].

## 5.5. Results

Individual macaques exhibited significant repeatability in the reversal-learning accuracy across trials and session ($R = 0.01$, CI = [0.002, 0.021], $p < 0.001$). When adjusting for session the performances were still repeatable ($R = 0.02$, CI = [0.005, 0.03], $p < 0.001$). These results mean that macaques exhibited stable individual variations in their performances between trials and session, even when the confounding factor session is considered.

There was a significant main effect of the age of the subjects on the number of trials required to learn the rules ($\chi^2_1 = 4.526$, $N = 19$; $p = 0.03$, see electronic supplementary material 1, table S11; [81]), with a higher number of trials required to learn the rule as the subject gets older (table 4 and figure 4). There was also a significant main effect of the rule on the number of trials required to reach the criterion ($\chi^2_1 = 32.989$, $N = 19$, $p < 0.001$, see electronic supplementary material 1, table S11; [81], the subjects needed less trials to learn the reversed rule (table 4). There was no interaction between the age and the rule ($\chi^2_1 = 0.101$, $N = 19$, $p = 0.751$, see electronic supplementary material 1, table S11; [81]). Other factors had no effect on the models (see electronic supplementary material 1, S11). We found no rank effect in females ($\chi^2_1 = 0.106$, $N = 9$, $p = 0.75$), nor a significant main effect of the age on the number of trials required to learn the rules ($\chi^2_1 = 1.58$, $N = 9$, $p = 0.21$ see electronic supplementary material 1, table S12; [81]). We also found that the subjects needed less trials to learn the reversed rule ($\chi^2_1 = 264.1$, $N = 9$, $p < 0.001$, see electronic supplementary material 1, table S12 and S13; [81]).

Regarding the accuracy on a trial, the session had a main significant effect ($\chi^2_1 = 8.368$, $N = 19$, $p = 0.004$; see electronic supplementary material 1, S14; [81]), with a better accuracy as the number of the session increased (see electronic supplementary material 1, S15, [81]). The age had only a tendency to

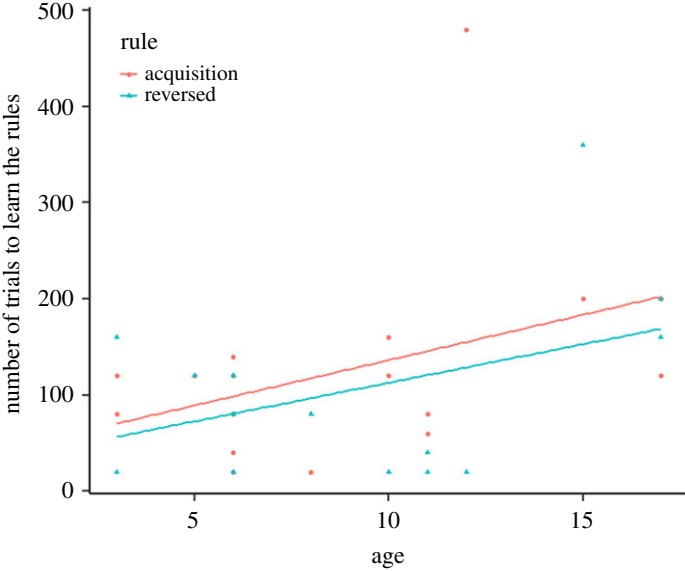

**Figure 4.** Number of trials to learn each rule (reach the criterion of success for the acquisition phase and the reversal phase) in function of the age in the reversal learning task.

**Table 4.** Results of the GLMM for the number of trials to learn the rules for the reversal-learning task. Explanatory variables were individual characteristics (sex and age) and type of rule (acquisition or reversed rule). All full models included the individual ID as a random factor. The estimates (representing the change in the dependent variable relative to the baseline category of each predictor variable), standard error, $t$-value and $p$-value using ML method. The variables age and rule had a significant main effect on the models. $*p < 0.05$, $***p < 0.001$.

| predictor number of trials | estimate | s.e. | z-value | Pr(>|t|) |
|---|---|---|---|---|
| (intercept) | 4.421 | 0.370 | 11.937 | $2.74 \times 10^{-50}$ |
| rule (reversed) | −0.183 | 0.032 | −5.739 | $9.87 \times 10^{-9}***$ |
| sex (male) | −0.197 | 0.266 | −0.739 | 0.511 |
| age | 0.064 | 0.033 | 1.939 | $0.018*$ |

have a significant effect when compared with the optimum model ($\chi^2_1 = 3.460$, $N = 19$, $p = 0.06$, see electronic supplementary material 1, S14; [81]). The other explanatory variables had no main effect on the models (see electronic supplementary material 1, S14; [81]). We found no rank effect in females ($\chi^2_1 = 0.252$, $N = 9$, $p = 0.62$), but we did find the same significant main effect of session ($\chi^2_1 = 6.041$, $N = 9$, $p = 0.01$; see electronic supplementary material 1, S16 and S17; [81]).

Regarding the perseveration error, the number of taps on the wrong stimulus, the trial ($\log_{10}$ of the number of taps, $\chi^2_1 = 5.999$, $N = 19$, $p < 0.05$; see electronic supplementary material 1, S18; [81]), the type of rule had a main significant effect of the number of taps ($\log_{10}$ of the number of taps, $\chi^2_1 = 9.042$, $N = 19$, $p < 0.01$; see electronic supplementary material 1, S18; [81]). Trial ($\log_{10}$ of the number of taps, $\chi^2_1 = 5.99$, $N = 19$, $p < 0.001$) and session had also a main significant effect ($\log_{10}$ of the number of taps, $\chi^2_1 = 13.819$, $N = 19$, $p < 0.001$; see electronic supplementary material 1, S18; [81]). The macaques made a lower number of taps on the wrong stimulus as trial and session increased (see electronic supplementary material 1, S19; [81]) and more taps on the wrong stimulus for the reversed rule compared with the acquisition rule. The other explanatory variables had no effect when compared with optimum level (see electronic supplementary material 1, S18; [81]). We found no rank effect in females ($\chi^2_1 = 2.919$, $N = 9$, $p = 0.09$), but we did also find the same significant main effect of the trial ($\log_{10}$ of the number of taps, $\chi^2_1 = 5.549$, $N = 19$, $p < 0.05$; see electronic supplementary material 1, S20 and S21; [81]) and session number in females ($\log_{10}$ of the number of taps, $\chi^2_1 = 13.779$, $N = 19$, $p < 0.001$; see electronic supplementary material 1, S20 and S21; [81]).

Regarding the response latency on a successful trial, the rule had a main significant effect (for log(of response latency), $\chi^2_1 = 7.168$, $N = 19$, $p < 0.001$; see electronic supplementary material 1, S22; [81]) with

a longer response latency for the reversed rule compared with the acquisition rule (see electronic supplementary material, S23; [81]). The session had also a main significant effect (for log(of response latency), $\chi^2_1 = 7.168$, $N = 19$, $p < 0.05$, see electronic supplementary material 1, S22; [81]), with longer response latency as session increased (see electronic supplementary material 1, S23; [81]). The other explanatory variables had no main effect on the models (see electronic supplementary material 1, S22; [81]). We found no rank effect in females ($\chi^2_1 = 0.254$, $N = 9$, $p = 0.87$), but we did find that their response latency was longer for the reversed rule (log(of response latency), $\chi^2_1 = 7.223$, $N = 9$, $p < 0.05$; see electronic supplementary material 1, S24 and S25; [81]).

## 5.6. Discussion

For the reversal-learning task, we found individual variation in the inhibition of a cognitive set. Surprisingly, older individuals had worse cognitive performances as they needed more trials to understand the rules.

We first found inter-individual variation in the inhibitory control performance across trials and sessions even when considering confounding factors. This indicates that some individuals were consistently better in the reversal-learning task than others. This variability in performance was also found in a study in dogs using the A-not-B task (a simplified reversal-learning task), which found that more than half of the dogs experienced difficulties in this task [46]. This result indicates that this task was sensitive enough to reveal individual variations.

We then demonstrated that the subjects in general needed less trial to learn the reversed rule than the first rule. It could be explained by the subjects making a transfer of learning between the first and the second rule, as they understand the general principle of the task.

We also demonstrated that middle-aged individuals needed more trials to learn the rules than younger ones, both in the acquisition phase and the reversion phase. However, there was not a clear difference in their accuracy on each trial. This result could mean that middle-aged individuals have a comparable accuracy to perform the task, but they stay for a longer time below the criterion of success as they might need more time to understand and memorize the rule. Nonetheless, the accuracy increased within and between sessions, showing that all the macaques can learn the rules. These results are at odds with other studies in primates showing an improvement of physical cognitive abilities as they get more experienced [42,43]. Similarly, brown lemurs used previous knowledge to master a novel reverse-reward task through generalization [49]. This experience difference could be explained by the possible contribution of cognitive flexibility (also part of executive functions) diverse from inhibitory control [19]. With younger individuals being more flexible which could help them to learn the rules.

This impairment in cognitive abilities in middle-aged macaques could also have been due to age-related decline. However, the older individuals of our sample were 17 years old which can be considered as middle aged in rhesus macaques [114]. Still, a study in chimpanzees showed age-related decline in cognitive flexibility that is already observed at middle age [115]. In a study with rhesus macaques, researchers compare executive function between young adults (5 to 9 years old), middle-aged macaques (12 to 19 years old) and aged macaques (20 to 30 years old). They found an impairment in the cognitive flexibility task in aged macaques but surprisingly also in middle-aged macaques [116]. Recent data in humans showed that age-related changes in cognition, particularly in executive function, may occur much earlier than expected [117,118]. When looking at the Wisconsin card sorting task, a benchmark task to study cognitive flexibility, not only individuals of advanced age are less efficient at this task, but so are those of middle age [118]. Executive function may be the earliest domain of cognitive function to show impairment in normal ageing [116,117]. Deficits in executive functions occur as an age as early as 40 [117], comparable with the middle-aged monkeys in our study.

If we consider that the middle-aged macaques of our study are also showing impairment in executive functions, our results could be similar to studies showing age-related disfunction in reversal-learning task. For instance, studies in humans showed age-related decline in reversal-learning performances [118,119]. Similarly, older dogs [120,121], older rats [122] or older non-human primates [123–126] were less flexible than younger ones. As in our study, Tapp *et al.* [120] demonstrated that old dogs were impaired on both the initial learning phase and the reversal phase. However, in Japanese macaques [124], the impairment was exhibited in the learning phase only, with a difficulty to associate the stimulus to the reward. This acquisition deficit was explained as a deficit in attending the relevant stimulus. On the contrary, Kumpan *et al.* [125] and Tsuchida *et al.* [126] showed that this age impairment was only found in the reversal phase. Tsuchida *et al.* [126] explained that old monkey's deficit was due to an impairment of understanding the

association between stimulus and reward. An explanation for the age impairment we found in both phases could be that our old subjects potentially encountered both types of difficulties during the task.

An explanation proposed for the age impairment in inhibitory control is a decline in the functioning of a region critical for inhibitory control, the prefrontal cortex [24,127] and particularly a decrease in the functioning of the dopamine system in this region [24]. Interestingly, it has been proposed that many aspects of age-related cognitive deficits, such as distractibility and impaired memory could be due to a failure in inhibitory mechanism and particularly an inability to inhibit irrelevant information [24,128]. In humans, there is evidence of a U-shaped model of the inhibitory control performances, as this ability develops slowly during childhood and decreases with age [128,129], it would be interesting to test immature macaques of less than 3 years old and compare their performances with adult and senior macaques.

Besides, we did not find a perseverative error pattern on the wrong stimulus in older individuals. This absence of perseverative error was also found in Japanese macaques [126]. However, in studies in dogs and non-human primates, old individuals persevered to respond to the wrong stimulus [67,120,124]. In our study, as trials and sessions increased the subjects made less perseverative error maybe due to a decrease in their motivation and arousal. This change in motivation could potentially have masked the age difference in perseverative error. Besides, we found that the subjects made more taps as the trials and sessions progressed, maybe due to a loss of patience and attention. Furthermore, they did more taps on the wrong stimulus on the reversed rule compared to the acquisition rule, they probably encountered more difficulty to learn the second rule as it contradicted with their initial learning.

Regarding the response latency on a successful trial, we demonstrated that it took more time for our subject in the reversing phase probably because of the interaction effect of the first rule in the learning of a new one. The females were also taking more time to respond to the stimulus, confirming their tendency to be less impulsive [52,53].

# 6. General discussion

The aim of this study was threefold: (i) to investigate individual variability in inhibitory control skills; (ii) to replicate, depending on the task, the most common effects of individual and group determinants on inhibitory control performances; and (iii) to demonstrate the effect of these factors on the three main components of inhibitory control. We first demonstrated individual variation in all components of inhibitory control. We then found an effect of sex and age on specific inhibitory control tasks. Males were more distracted by pictures than females, and all the subjects were particularly distracted by the threatening stimuli. Middle-aged monkeys had impaired cognitive performances in the reversal-learning task. We did not find an effect of the rank of the individuals in any of the inhibitory control tasks. Finally, we found that sex and age affected the inhibitory control performances differently across the tasks. These results thus could give a new insight into the multifaceted structure of inhibitory control.

Firstly, we found that macaques showed consistent individual variation in all three components of inhibitory control. These findings confirm individual differences in inhibition of action found in guppies using a variant of the detour-reaching task [30] and in pheasants using a response inhibition task [33]. Montalbano *et al.* [32] demonstrated individual differences in guppies and showed positive covariation between two different measures of inhibition of action (the tube task and the cylinder task). However, another study in pheasants did not demonstrate a stable individual variation using common tasks of inhibitory control (detour-reaching task and reversal-learning task [35]). Nonetheless, it seems important to point out that the validity and reliability of the detour-reaching task, the cylinder task and the A-not-B has been recently challenged [23,27,44,47,48]. These contradictory results between studies could then be explained by this lack of repeatability of measurements. In our study, we used reliable and valid tasks of inhibitory control [73] and we systematically examine individual differences in all three components of this ability. Before considering large comparative studies, further studies should therefore first systematically investigate individual differences in inhibitory control using tasks previously tested for reliability and validity.

According to a growing number of authors, it is crucial to understand individual variability in a cognitive process to better apprehend its evolution [23,24,26,29]. As Darwin pointed out, individual variations 'afford materials for natural selection to act on' [130]. Indeed, investigating individual differences is essential to our comprehension of how natural selection sifts individual difference leading to evolutionary changes [24]. In a large comparative study, using over 36 species [14], the

authors made claims about the evolutionary underpinnings of inhibitory control, yet this study was not considering individual differences. In our study, we demonstrated this necessary evolutionary condition of individual variation for the three main components of inhibitory control. Thus, the tasks used in our study, revealing inter-individual variation, could then be used in further comparative studies to look at the evolutionary underpinnings of this ability.

We also found that the individual characteristics (age and sex) influence the different components of inhibitory control inconsistently. Age only influenced performances in the inhibition of cognition, and sex only influenced performances in the inhibition of a distraction. This inconsistency of the effect of the different factors influencing inhibitory control can be found in other studies. For example, Bray et al. [67] found that the factor age only influenced dog's performances in the inhibition of action (cylinder task) but not the inhibition of cognition (A-not-B task). However, Vernouillet et al. [46] found an effect of age on the performance only in a modified version of the A-not-B task with barrier (mixing inhibition of action and cognition) but not in the inhibition of cognition (A-not-B task) nor action (cylinder task). Inconsistent performances and contradictory effect of the influencing factors between the cylinder task, the detour task and the A-not-B task could be explain by the task impurity problem, common in animal research. When building a task, other cognitive requirements (memory, learning, physical understanding, etc.) might be implicated in the variation of performances but without being directly relevant to the targeted function [27,75]. Besides, observed variation may reflect individual differences (e.g. motivation or personality) or idiosyncratic task requirements (e.g. context of the task or salience of the reward) with only a small proportion of the variation capturing changes in the studied process [23]. For example, guppies having bolder personalities showed greater inhibitory control abilities in the tube task [87]. This task impurity problem is even more important with the cylinder task, the detour task and the A-not-B task, of which validity, as seen previously, has been recently challenged [23,27,44,47,48]. Factors such as context [44,46,50,51,67] can dramatically influence subjects' performance. In our study, we used a valid battery of inhibitory control task [73], and we minimized as much as possible the task impurity problem. For instance, all tasks were performed in the same context (on a touchscreen, training phases, using the same stimuli for the targets), with subjects with similar engagement (prior habituation, feeding ad libitum, reward of the same salience etc.). We used basic rules for all the tasks, for the inhibition of cognition and action, equivalent cognitive load was engaged in both the tasks as the subjects had only to memorize that one symbol was rewarded and the other one not rewarded. In our study, the inconsistent effect of age and sex on the different components of inhibitory control seems to point forward a multi-faceted structure of inhibitory control, divided in independent components, as proposed by several authors [6,7,74,75]. In another study [73], we also looked at the repeatability of performances between tasks. We found the consistency of performance between the inhibition of a distraction and the inhibition of an action; this was not found for inhibition of a pre-learned rule which strengthens the hypothesis of a multi-faceted structure of inhibitory control.

We did not find any effect of the social rank on inhibitory control performance across all tasks. This could be due to a small sample size as we tested only four females from lower rank. Inconsistent effect of age and sex could also be due to our small sample size. However, studies looking specifically at individual differences in guppies had a sample size going from 22 to 28 individuals [30–32,37,101]. A low sample size, a common limitation when working with primates, might have decreased the power of our analysis [112]. Moreover, our results are potentially only representative of one sample of one population of captive rhesus macaques. Before considering interspecies comparison, it would be interesting to first replicate this task battery in another population of captive rhesus macaques. Second, a bigger challenge would be to adapt these tasks to a wild population of rhesus macaques. Indeed, compared with captive animals, free-living population have radically different developmental trajectories and potentially important differences in inhibitory control [24]. Recent studies made a remarkable effort to adapt inhibitory control tasks in wild populations of hyenas [36], robins [131], vervet monkeys [125] and Australian magpies [38]. We hope for a large-scale collaborative project across laboratories or field sites to increase sample size and diversity in animal cognition experiments. Standardized experimental protocols and online data repositories shared between laboratories would be ideal to improve statistical rigour (e.g. the ManyPrimates project [132]).

In our study, we did not consider important components of the social environment which could explain variation in inhibitory control. According to the social intelligence hypothesis [61,62], the more complex a social life is, the more cognitive performances of a higher order, such as inhibitory control, might be selected (e.g. [63]). For instance, the size of a social group has been used as a proxy for social complexity. Living in larger groups may be more cognitively demanding due to an increase in the numbers of differentiated relationships and interactions between groupmates [133,134]. It has

been demonstrated with the cylinder task in spotted hyenas that developing in a larger group generates better inhibitory control skills [36]. This result was also found in Australian magpies with the cylinder task and the reversal-learning task [38]. However, MacLean *et al.* [135] using the cylinder task with six species of primates did not find an effect of the group size on their performances. It would be interesting to include in our analysis the size of the group in which our subjects developed. If developing in a larger group size (considered as a proxy for social complexity) was associated with better inhibitory control skills, then the social intelligence hypothesis could be corroborated.

Moreover, it has also been suggested that one route by which social cognition can evolve is through selection on social tolerance [136]. Tolerant social styles feature higher reconciliation rates, fewer conflicts and more relaxed social relationships than despotic ones [69]. It is possible that individuals living in a more tolerant social context might experience more diverse and complex social interactions and consequently would have better inhibitory control skills. For example, in Joly *et al.* [137] four macaque species with different degrees of social tolerance underwent an inhibitory control task. More tolerant species outperformed less tolerant species in this task but only one task of inhibition of action was conducted. It would be interesting to try to replicate the previous findings in more than one task of inhibitory control, in several macaque species, to determine if more tolerant species outperform the less tolerant ones in all the components of inhibitory control.

To summarize, we first found that individuals showed variability in inhibitory control performances. We also demonstrated that the age and the sex of an individual act distinctly on the different components of inhibitory control. These results are in favour of the multi-faceted structure of inhibitory control. Thus, further studies on individual variation in inhibitory control performances can provide the basis for a novel and powerful approach to understanding the evolution of this ability.

Ethics. This study was approved by the Animal Welfare and Ethical Review Body of the University of Portsmouth, AWERB no. 4015B and by the MRC-CFM's Animal Welfare and Ethical Review Body ARWEB no. CFM2019E002. This study was part of the Macaque Cognition Project, University of Portsmouth, and was approved by the Animal Welfare and Ethical Review Body of the University of Portsmouth, AWERB no. 4015B and by the MRC-CFM's Animal Welfare and Ethical Review Body, AWERB no. CFM2019E002.

Data accessibility. The Matlab codes and stimuli for every task, the datasets generated during the current study, R scripts used for the analysis and electronic supplementary material 1 and 2 are stored in Github: https://github.com/Psychology-inhibitory-control/individual-differences/tree/v1.0.0 and have been archived within the Zenodo repository: https://zenodo.org/badge/DOI/10.5281/zenodo.5577452.svg [81].

Authors' contributions. L.L. and M.J. conceived and designed the study. L.L. collected the data, carried out the statistical analyses, wrote and drafted the manuscript. J.M. gave statistical advice. B.M.W., J.M. and M.J. helped draft the manuscript. All authors gave final approval for publication.

Competing interests. We have no competing interests.

Funding. This study was supported by the University of Portsmouth Faculty Bursary, the Primate Society of Great Britain research grant and the International Primatological research grant.

Acknowledgements. We are grateful to Margot Moniot and Elen Stanton for their help in collecting the data. Many thanks to Dr Claire Witham and all the caretakers from the MRC, UK for their help in coordinating *in situ* the collection of the data. Thank you to Dr Claire Witham for letting us use her pictures of the macaques. Thank you to Florent Le Moël and Alexandre Montlibert for their help in creating the Matlab scripts.

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
