## [Peer Review File · Royal Society Open Science]

Review History

RSOS-202102.R0 (Original submission)

Review form: Reviewer 1

Is the manuscript scientifically sound in its present form?

Yes

Are the interpretations and conclusions justified by the results?

Yes

Is the language acceptable?

Yes

Do you have any ethical concerns with this paper?

No

Have you any concerns about statistical analyses in this paper?

No

Recommendation?

Accept with minor revision (please list in comments)

Comments to the Author(s)

The MS RSOS202102 reports a set of experiments aimed to investigate individual differences in inhibition in macaques. I found this study very interesting, with the important novelty of trying to assess different aspects of a cognitive function and different predictors of individual differences simultaneously. While this approach is common in psychology research in humans, it is quite rare in other species. For this reason, I feel that this paper would be an important contribution for research in the field. Below I suggest changes that should improve the presentation of the study.

Detailed comments

Abstract

L28 It is not clear the meaning of "the demographic factors did not influence the main components of inhibitory control". Please provide more details or re-phrase.

Introduction

The introduction was clear and easy to follow. However, often it failed to acknowledge prior research in other animal species. I am confident that this issue can be fixed by discussing opportune references. I have suggested some studies performed in my lab and from other colleagues that might be of interest for this point.

L11-15 I believe that this is a novel and important approach for animal behaviour studies. If possible, it would be beneficial to underlie it in the abstract.

L32-34 This sentence is a bit misleading because in the same species there is evidence of individual differences for several other IC tasks:

Lucon-Xiccato, T., Montalbano, G., Dadda, M., & Bertolucci, C. (2020). Lateralization correlates with individual differences in inhibitory control in zebrafish. *Biology Letters*, 16(8), 20200296.

Montalbano, G., Bertolucci, C., & Lucon-Xiccato, T. (2020). Measures of inhibitory control correlate between different tasks but do not predict problem-solving success in a fish, *Poecilia reticulata*. *Intelligence*, 82, 101486.

L38 There is an example in fish and one example in birds. I would recommend to add citations to previous studies on individual differences in mammals, including primates, as this study is pertinent to a primate species.

For example:

Brucks, D., Marshall-Pescini, S., Wallis, L. J., Huber, L., & Range, F. (2017). Measures of dogs' inhibitory control abilities do not correlate across tasks. *Frontiers in Psychology*, 8, 849.

Beran, M. J., & Hopkins, W. D. (2018). Self-control in chimpanzees relates to general intelligence. *Current Biology*, 28, 574-579.

L60 Perhaps it would be useful to cite this work finding the same effect in birds

Kabadayi, C., Jacobs, I., & Osvath, M. (2017). The development of motor self-regulation in ravens. *Frontiers in psychology*, 8, 2100.

L74-75 This sentence is a bit misleading. I think that there are evidence of sex differences in IC in animals, but not relating to distraction inhibition. Please rephrase this sentence and add references such as this paper cited below:

Lucon-Xiccato, T., Bisazza, A., & Bertolucci, C. (2020). Guppies show sex and individual differences in the ability to inhibit behaviour. *Animal Cognition*, 1-9.

Methods

L150 The sample size seems generally OK compared to previous studies. However, I am not sure about the power to detect age differences. As other readers may have my same concern, I suggest to try to address this point or make a short comment in discussion.

L160-161 From my understanding, the subjects were naïve of cognitive tasks. I suggest to make clear this point as very often primates used in cognitive studies have a quite long experience with cognitive experiments, and this can affect individual differences in performance.

Experiment 1

L247 It would be helpful to use the terms Experiment 1, Experiment 2 etc also in introduction.

L271 Is there evidence that this species perceives the pictures as real conspecifics? If so, it would be nice to have a reference to strength the validity of the paradigm.

L313 This is the type of previous experimental experience that I asked to address above.

L319 Was there an improvement (learning) over trials and sessions?

L336 Please give more details about these effects. Are they a form of learning to solve the task more efficiently with practice?

There are several small spacing errors in the result section of this experiment (eg. L346).

L374 The example of guppies is apparently a form of inhibition of action. If correct, this may be explicitly discussed (maybe also in the discussion of exp. 2).

Experiment 3

I agree that the reversal learning task requires some form of inhibition. However, according to many authors it also requires flexibility, which is often considered an executive function diverse from inhibitory control (Diamond, 2003). I suggest to acknowledge the possible contribution of flexibility in the results of reversal learning task.

General discussion

Some of the subjects performed more than one experiment. This theoretically allows the authors to test for correlations between task performances, which provides a measure of inter task individual differences. Is it possible to add this analysis and use it for the general discussion?

L654 The study by Montalbano et al. (reported above) showing positive covariation between two different measures of inhibitory control can be useful here.

L664 It has been recently proposed that similar individual differences might be related to personality. I am wondering if this should be addressed here.

Review form: Reviewer 2

Is the manuscript scientifically sound in its present form?

No

Are the interpretations and conclusions justified by the results?

No

Is the language acceptable?

Yes

Do you have any ethical concerns with this paper?

No

Have you any concerns about statistical analyses in this paper?

Yes

Recommendation?

Major revision is needed (please make suggestions in comments)

Comments to the Author(s)

This manuscript examines three components of executive function in 21 rhesus macaques using cognitive tasks implemented with touch screen computers. The main goals are to examine different sources of individual variation as well as examined the structure of cognitive performance. The paper makes a good case for the importance of studying individual variation, something that has been a major focus of work in human cognitive sciences but less often a focus of comparative studies. While documenting individual variation in valuable, most of the time the real puzzle is explaining it. Unfortunately, I am not convinced the current paper appropriately tackles that second component, and I am also not sure the methodological approach is ideal for capturing (repeatable) individual variation as described. In addition, I have several clarification questions regarding what the specific procedures and analyses were.

1. Aims, hypotheses and predictions

The introduction mentions multiple potential sources of variation (pg. 2-4), but it seems like the devil is in the details concerning which kinds of self-control tasks are being discussed. In particular, the literature cited here deals with a fairly wide range of decision-making and executive function tasks, some of which seem quite related to the current tasks (e.g., age effects in reversal learning) but some of which are not as clearly linked (e.g., sex differences in sensation seeking or risk-taking), and which further vary by species. This makes the subsequent hypotheses for sources of individual variation in the specific tasks appear not appear well-explained.

For example, why is a sex difference (but not an age or rank difference) predicted in the Stroop task? This hypothesis shapes the analytic choices, as age and rank are not even examined in this study. In contrast, sex, age, and rank are predicted to impact performance on the go/no-go task (based on previous studies with various species, not a-priori theoretical hypotheses), leading to a different set of models even though the same subjects (mostly) are tested. Finally, only age is examined in the reversal learning task following the same kinds of logic. Overall, this makes the analysis procedures for each task appear somewhat arbitrary.

Finally, one of the major stated goals of the paper is to examine the structure of inhibitory control. First off, I would point out that typically in psychology such a question would be addressed with a factor analysis (e.g., see Miyake et al 2000 Cognitive Psychology for an examine from human executive functions, or MacLean et al 2017 Animal Behaviour for a similar approach from animal cognition). This allows looks at common versus shared variance across different measures to infer the latent structure of these cognitive processes. However, this study likely has too few individuals (and too few independent task measures) to do this properly. Instead, the paper seems to take the approach of seeing if the same age-, sex-, and rank- effects account for performance across the different contexts. However, currently the manuscript does not report clearly parallel analyses testing for all of these effects in each task (as described above) which means this is not clearly assessed. More generally, I think there is a real theoretical question of

whether looking at shared age-, sex-, or rank- effects across tasks is actually good evidence relevant to understanding the structure of cognitive abilities in the first place. While I think the question is interesting, I would say this is the weakest output of the paper and I would urge either taking a more well-understood approach from cognitive science to answer this question or else providing more clear justification for why this approach is valid.

2. Study population and sample size

While I appreciate the difficulty in acquiring a larger sample size of animals, 21 individuals is a small sample size for testing questions about individual differences. Indeed, this kind of sample size limitation is exactly one reason why there are so few studies of individual variation in nonhuman cognition. Nonetheless, I think it would be important to better justify how this sample size is appropriate for the questions set up here. For example, are there past studies showing age, sex, or rank effects in rhesus monkeys with comparable sample sizes? Could the authors perform a power analysis?

I also have some questions about the monkeys. It seems like the final sample of 21 was drawn from multiple groups – are these of species typical size; do they include mixed sex or age individuals? Who do the subject monkeys live with? This kind of information about their social experiences is important for interpreting the findings about sex, age, and rank differences. In addition, little information about how the dominance rank information was acquired is provided. Was this just the opinion of staff, or stemming from systematic behavioural observations? As currently written it sounds like it may have been staff opinion. If so, how reliable is this metric? Since 15 of 21 final sample of subjects are reported to be high-ranking, this seems rather skewed.

3. Methods

One general comment is that the validation of the tasks used here appears to be unpublished (e.g., it is cited as Loyant et al., submitted), and I have several comments on whether these tasks are actually assessing what they aim to.

Stroop task: For example, it's not clear to me that Study 1 (Stroop task) is properly considered a Stroop task. The task used with monkeys involved touching a target red rectangle, and in different conditions various distractors (a neutral object, a neutral conspecific, or a threatening conspecific) also appear on the screen. The analyses examined whether these distractors influenced reaction time. Yet the human Stroop task is not focused on whether people are distracted by irrelevant extra stimuli, but on how a stimulus feature can distract processing another stimulus feature. For example, imagine the task is to report how many words appear on a screen, and then seeing a screen with a word printed three times. While it might be easy to correctly report "3" if the word printed three times is "cat," it is more challenging to report "3" when the actual word printed three times is 'four'. This conflict between different dimensions of the stimuli is key to the Stroop task, and the monkey task does not seem to have this element of needing to shift attention between different aspects of the stimuli.

Go/no go task: In the go/no-go task, it appears that monkeys completed 10 sessions of 20 trials each (pg. 19) but only the last 3 were analysed. It is unclear if these took place on the same day or not. One important consideration is whether the monkeys had actually learned the difference between the go and no-go options given that it appears they had to acquire this only through direct experience with choosing the options. Is there some other validation that the animals understood the task?

Finally, the specifics on trial, sessions, and testing days for the different tasks is a bit confusing and spread out over multiple sections. It would be helpful to consolidate this so the reader can have a clearer picture of what actually happened. For example, the general method indicates that the tasks were conducted in the same order for each subject and that once all the sessions of a task

were complete, the next task was conducted on the next testing day (page 8-9) which would imply that monkeys did each task on separate days and completed multiple “sessions” per task on the same day. Then, the Stroop task seems to have involved 3 sessions of 36 trials each, with each session split into 6 blocks (page 12). What is the significance of sessions versus blocks? Did the monkey get a pause? This matters both for understanding the actual procedure, as well as for interpreting the repeatability scores reported for the tasks as discussed below.

4. Analyses and results interpretation

I already discussed above how different analysis procedures are used on the different tasks such that the impact of age, sex, and rank are tested in the go/no-go task, but only sex is mentioned in the Stroop task and only age in the reversal task.

Another issue concerns the repeatability analyses. For example, it is reported that individual macaques exhibited significant repeatability in these tasks. But, how meaningful is this if all trials/blocks/sessions occurred on the same day? Repeatability in behavioural metrics, for example, generally look at whether there is a core “personality” dimension that predicts behaviour across contexts and over time (for example, Tkaczynski et al 2020 Royal Society Open Science). It seems like a comparable claim for stable individual differences in cognitive performance would require testing the same individual at least on different days. Assuming that cognitive performance can be influenced by both stable individual differences but also situational factors (what’s going on in the social group, the monkey’s emotional state, hunger, etc) these situational factors will be common across the testing sessions from the same day and thus could account for apparent repeatability in analyses.

Decision letter (RSOS-202102.R0)

Dear Miss Loyant

The Editors assigned to your paper RSOS-202102 "Age and sex individual differences reveal a multifaceted structure of inhibitory control in non-human primates" have made a decision based on their reading of the paper and any comments received from reviewers.

Regrettably, in view of the reports received, the manuscript has been rejected in its current form. However, a new manuscript may be submitted which takes into consideration these comments.

We invite you to respond to the comments supplied below and prepare a resubmission of your manuscript. Below the referees’ and Editors’ comments (where applicable) we provide additional requirements. We provide guidance below to help you prepare your revision.

Please note that resubmitting your manuscript does not guarantee eventual acceptance, and we do not generally allow multiple rounds of revision and resubmission, so we urge you to make every effort to fully address all of the comments at this stage. If deemed necessary by the Editors, your manuscript will be sent back to one or more of the original reviewers for assessment. If the original reviewers are not available, we may invite new reviewers.

Please resubmit your revised manuscript and required files (see below) no later than 03-Aug-2021. Note: the ScholarOne system will ‘lock’ if resubmission is attempted on or after this

deadline. If you do not think you will be able to meet this deadline, please contact the editorial office immediately.

Please note article processing charges apply to papers accepted for publication in Royal Society Open Science (<https://royalsocietypublishing.org/rsos/charges>). Charges will also apply to papers transferred to the journal from other Royal Society Publishing journals, as well as papers submitted as part of our collaboration with the Royal Society of Chemistry (<https://royalsocietypublishing.org/rsos/chemistry>). Fee waivers are available but must be requested when you submit your manuscript (<https://royalsocietypublishing.org/rsos/waivers>).

Thank you for submitting your manuscript to Royal Society Open Science and we look forward to receiving your resubmission. If you have any questions at all, please do not hesitate to get in touch.

on behalf of Dr Oliver Schülke (Associate Editor) and Kevin Padian (Subject Editor)
openscience@royalsociety.org

Associate Editor Comments to Author (Dr Oliver Schülke):

Associate Editor: 1

Comments to the Author:

Dear Dr. Loyant,

as the associate editor handling your submission I have now acquired comments from two expert reviewers. Based on their judgement and my own reading of the manuscript I have to ask you to thoroughly revise it before we can consider it for publication in RSOS. Both reviewers make excellent suggestions for improvement of the text. I will reject the manuscript in its current form for two reasons: 1) the investigation into age effects is motivated mainly by cognitive aging phenomena which is misleading in a study where the oldest individual is 17 years and pronounced effect of aging occurring only after that in the study species. So I suggest to frame the study of age as a predictor with experience effects instead. 2) Sex and rank effects cannot reliably be disentangled in a sample with only 2 low-ranking males. I suggest either restricting the investigation of rank effects to females or to omit sex as a predictor. I also would like to reiterate that a detailed description of methods applied for the construction of a dominance hierarchy along with a description of the data base (number of conflicts, spread across individuals and dyads) is required in a study that targets dominance effects. In addition to the issues raised by the reviewers I would also like to see the number of subjects participating in the study (21) to be mentioned more prominently instead of starting the paragraph with describing the composition of the group they live in (30). If you are able to address the critique in a satisfactory way and revise your manuscript accordingly, I will be happy to reconsider it for publication in RSOS.

With kind regards,
Oliver Schülke

Associate Editor (RSOS)

Reviewer comments to Author:

Reviewer: 1

Comments to the Author(s)

The MS RSOS202102 reports a set of experiments aimed to investigate individual differences in inhibition in macaques. I found this study very interesting, with the important novelty of trying to assess different aspects of a cognitive function and different predictors of individual differences simultaneously. While this approach is common in psychology research in humans, it is quite rare in other species. For this reason, I feel that this paper would be an important contribution for research in the field. Below I suggest changes that should improve the presentation of the study.

Detailed comments

Abstract

L28 It is not clear the meaning of “the demographic factors did not influence the main components of inhibitory control”. Please provide more details or re-phrase.

Introduction

The introduction was clear and easy to follow. However, often it failed to acknowledge prior research in other animal species. I am confident that this issue can be fixed by discussing opportune references. I have suggested some studies performed in my lab and from other colleagues that might be of interest for this point.

L11-15 I believe that this is a novel and important approach for animal behaviour studies. If possible, it would be beneficial to underlie it in the abstract.

L32-34 This sentence is a bit misleading because in the same species there is evidence of individual differences for several other IC tasks:

Lucon-Xiccato, T., Montalbano, G., Dadda, M., & Bertolucci, C. (2020). Lateralization correlates with individual differences in inhibitory control in zebrafish. *Biology Letters*, 16(8), 20200296.

Montalbano, G., Bertolucci, C., & Lucon-Xiccato, T. (2020). Measures of inhibitory control correlate between different tasks but do not predict problem-solving success in a fish, *Poecilia reticulata*. *Intelligence*, 82, 101486.

L38 There is an example in fish and one example in birds. I would recommend to add citations to previous studies on individual differences in mammals, including primates, as this study is pertinent to a primate species.

For example:

Bruks, D., Marshall-Pescini, S., Wallis, L. J., Huber, L., & Range, F. (2017). Measures of dogs' inhibitory control abilities do not correlate across tasks. *Frontiers in Psychology*, 8, 849.

Beran, M. J., & Hopkins, W. D. (2018). Self-control in chimpanzees relates to general intelligence. *Current Biology*, 28, 574-579.

L60 Perhaps it would be useful to cite this work finding the same effect in birds

Kabadayi, C., Jacobs, I., & Osvath, M. (2017). The development of motor self-regulation in ravens. *Frontiers in psychology*, 8, 2100.

L74-75 This sentence is a bit misleading. I think that there are evidence of sex differences in IC in animals, but not relating to distraction inhibition. Please rephrase this sentence and add references such as this paper cited below:

Lucon-Xiccato, T., Bisazza, A., & Bertolucci, C. (2020). Guppies show sex and individual differences in the ability to inhibit behaviour. *Animal Cognition*, 1-9.

Methods

L150 The sample size seems generally OK compared to previous studies. However, I am not sure about the power to detect age differences. As other readers may have my same concern, I suggest to try to address this point or make a short comment in discussion.

L160-161 From my understanding, the subjects were naïve of cognitive tasks. I suggest to make clear this point as very often primates used in cognitive studies have a quite long experience with cognitive experiments, and this can affect individual differences in performance.

Experiment 1

L247 It would be helpful to use the terms Experiment 1, Experiment 2 etc also in introduction.

L271 Is there evidence that this species perceives the pictures as real conspecifics? If so, it would be nice to have a reference to strength the validity of the paradigm.

L313 This is the type of previous experimental experience that I asked to address above.

L319 Was there an improvement (learning) over trials and sessions?

L336 Please give more details about these effects. Are they a form of learning to solve the task more efficiently with practice?

There are several small spacing errors in the result section of this experiment (eg. L346).

L374 The example of guppies is apparently a form of inhibition of action. If correct, this may be explicitly discussed (maybe also in the discussion of exp. 2).

Experiment 3

I agree that the reversal learning task requires some form of inhibition. However, according to many authors it also requires flexibility, which is often considered an executive function diverse from inhibitory control (Diamond, 2003). I suggest to acknowledge the possible contribution of flexibility in the results of reversal learning task.

General discussion

Some of the subjects performed more than one experiment. This theoretically allows the authors to test for correlations between task performances, which provides a measure of inter task individual differences. Is it possible to add this analysis and use it for the general discussion?

L654 The study by Montalbano et al. (reported above) showing positive covariation between two different measures of inhibitory control can be useful here.

L664 It has been recently proposed that similar individual differences might be related to personality. I am wondering if this should be addressed here.

Reviewer: 2

Comments to the Author(s)

This manuscript examines three components of executive function in 21 rhesus macaques using cognitive tasks implemented with touch screen computers. The main goals are to examine different sources of individual variation as well as examined the structure of cognitive performance. The paper makes a good case for the importance of studying individual variation, something that has been a major focus of work in human cognitive sciences but less often a focus of comparative studies. While documenting individual variation in valuable, most of the time the

real puzzle is explaining it. Unfortunately, I am not convinced the current paper appropriately tackles that second component, and I am also not sure the methodological approach is ideal for capturing (repeatable) individual variation as described. In addition, I have several clarification questions regarding what the specific procedures and analyses were.

1. Aims, hypotheses and predictions

The introduction mentions multiple potential sources of variation (pg. 2-4), but it seems like the devil is in the details concerning which kinds of self-control tasks are being discussed. In particular, the literature cited here deals with a fairly wide range of decision-making and executive function tasks, some of which seem quite related to the current tasks (e.g., age effects in reversal learning) but some of which are not as clearly linked (e.g., sex differences in sensation seeking or risk-taking), and which further vary by species. This makes the subsequent hypotheses for sources of individual variation in the specific tasks appear not appear well-explained.

For example, why is a sex difference (but not an age or rank difference) predicted in the Stroop task? This hypothesis shapes the analytic choices, as age and rank are not even examined in this study. In contrast, sex, age, and rank are predicted to impact performance on the go/no-go task (based on previous studies with various species, not a-priori theoretical hypotheses), leading to a different set of models even though the same subjects (mostly) are tested. Finally, only age is examined in the reversal learning task following the same kinds of logic. Overall, this makes the analysis procedures for each task appear somewhat arbitrary.

Finally, one of the major stated goals of the paper is to examine the structure of inhibitory control. First off, I would point out that typically in psychology such a question would be addressed with a factor analysis (e.g., see Miyake et al 2000 *Cognitive Psychology* for an examine from human executive functions, or MacLean et al 2017 *Animal Behaviour* for a similar approach from animal cognition). This allows looks at common versus shared variance across different measures to infer the latent structure of these cognitive processes. However, this study likely has too few individuals (and too few independent task measures) to do this properly. Instead, the paper seems to take the approach of seeing if the same age-, sex-, and rank- effects account for performance across the different contexts. However, currently the manuscript does not report clearly parallel analyses testing for all of these effects in each task (as described above) which means this is not clearly assessed. More generally, I think there is a real theoretical question of whether looking at shared age-, sex-, or rank- effects across tasks is actually good evidence relevant to understanding the structure of cognitive abilities in the first place. While I think the question is interesting, I would say this is the weakest output of the paper and I would urge either taking a more well-understood approach from cognitive science to answer this question or else providing more clear justification for why this approach is valid.

2. Study population and sample size

While I appreciate the difficulty in acquiring a larger sample size of animals, 21 individuals is a small sample size for testing questions about individual differences. Indeed, this kind of sample size limitation is exactly one reason why there are so few studies of individual variation in nonhuman cognition. Nonetheless, I think it would be important to better justify how this sample size is appropriate for the questions set up here. For example, are there past studies showing age, sex, or rank effects in rhesus monkeys with comparable sample sizes? Could the authors perform a power analysis?

I also have some questions about the monkeys. it seems like the final sample of 21 was drawn from multiple groups – are these of species typical size; do they include mixed sex or age individuals? Who do the subject monkeys live with? This kind of information about their social experiences is important for interpreting the findings about sex, age, and rank differences. In addition, little information about how the dominance rank information was acquired is provided.

Was this just the opinion of staff, or stemming from systematic behavioural observations? As currently written it sounds like it may have been staff opinion. If so, how reliable is this metric? Since 15 of 21 final sample of subjects are reported to be high-ranking, this seems rather skewed.

3. Methods

One general comment is that the validation of the tasks used here appears to be unpublished (e.g., it is cited as Loyant et al., submitted), and I have several comments on whether these tasks are actually assessing what they aim to.

Stroop task: For example, it's not clear to me that Study 1 (Stroop task) is properly considered a Stroop task. The task used with monkeys involved touching a target red rectangle, and in different conditions various distractors (a neutral object, a neutral conspecific, or a threatening conspecific) also appear on the screen. The analyses examined whether these distractors influenced reaction time. Yet the human Stroop task is not focused on whether people are distracted by irrelevant extra stimuli, but on how a stimulus feature can distract processing another stimulus feature. For example, imagine the task is to report how many words appear on a screen, and then seeing a screen with a word printed three times. While it might be easy to correctly report "3" if the word printed three times is "cat," it is more challenging to report "3" when the actual word printed three times is 'four'. This conflict between different dimensions of the stimuli is key to the Stroop task, and the monkey task does not seem to have this element of needing to shift attention between different aspects of the stimuli.

Go/no go task: In the go/no-go task, it appears that monkeys completed 10 sessions of 20 trials each (pg. 19) but only the last 3 were analysed. It is unclear if these took place on the same day or not. One important consideration is whether the monkeys had actually learned the difference between the go and no-go options given that it appears they had to acquire this only through direct experience with choosing the options. Is there some other validation that the animals understood the task?

Finally, the specifics on trial, sessions, and testing days for the different tasks is a bit confusing and spread out over multiple sections. It would be helpful to consolidate this so the reader can have a clearer picture of what actually happened. For example, the general method indicates that the tasks were conducted in the same order for each subject and that once all the sessions of a task were complete, the next task was conducted on the next testing day (page 8-9) which would imply that monkeys did each task on separate days and completed multiple "sessions" per task on the same day. Then, the Stroop task seems to have involved 3 sessions of 36 trials each, with each session split into 6 blocks (page 12). What is the significance of sessions versus blocks? Did the monkey get a pause? This matters both for understanding the actual procedure, as well as for interpreting the repeatability scores reported for the tasks as discussed below.

4. Analyses and results interpretation

I already discussed above how different analysis procedures are used on the different tasks such that the impact of age, sex, and rank are tested in the go/no-go task, but only sex is mentioned in the Stroop task and only age in the reversal task.

Another issue concerns the repeatability analyses. For example, it is reported that individual macaques exhibited significant repeatability in these tasks. But, how meaningful is this if all trials/blocks/sessions occurred on the same day? Repeatability in behavioural metrics, for example, generally look at whether there is a core "personality" dimension that predicts behaviour across contexts and over time (for example, Tkaczynski et al 2020 Royal Society Open Science). It seems like a comparable claim for stable individual differences in cognitive performance would require testing the same individual at least on different days. Assuming that cognitive performance can be influenced by both stable individual differences but also situational

factors (what's going on in the social group, the monkey's emotional state, hunger, etc) these situational factors will be common across the testing sessions from the same day and thus could account for apparent repeatability in analyses.

===PREPARING YOUR MANUSCRIPT===

===PREPARING YOUR REVISION IN SCHOLARONE===

<https://royalsociety.org/journals/authors/author-guidelines/#supplementary-material> to include a suitable title and informative caption. An example of appropriate titling and captioning may be found at https://figshare.com/articles/Table_S2_from_Is_there_a_trade-off_between_peak_performance_and_performance_breadth_across_temperatures_for_aerobic_sc_ope_in_teleost_fishes_/3843624.

Author's Response to Decision Letter for (RSOS-202102.R0)

See Appendix A.

Decision letter (RSOS-211564.R0)

Dear Miss Loyant,

I am pleased to inform you that your manuscript entitled "Heterogeneity of performances in several inhibitory control tasks: male rhesus macaques are more easily distracted than females" is now accepted for publication in Royal Society Open Science.

on behalf of Dr Oliver Schülke (Associate Editor) and Kevin Padian (Subject Editor)
openscience@royalsociety.org

Appendix A

Dear Editor,

Many thanks for your comments on our manuscript. Please find attached the second revised version with tracked changes, our replies to each of the questions (see below) as well as a “clean” version of the manuscript. We revised the paper according to your and the reviewers’ questions and comments. We have improved the text according to all the raised issues. We hope that this version is now acceptable for publication in Royal Society of Open Science.

Kind regards

Responses to comments:

Editor comments: Dr Oliver Schülke

as the associate editor handling your submission I have now acquired comments from two expert reviewers. Based on their judgement and my own reading of the manuscript I have to ask you to thoroughly revise it before we can consider it for publication in RSOS. Both reviewers make excellent suggestions for improvement of the text. I will reject the manuscript in its current form for two reasons: 1) the investigation into age effects is motivated mainly by cognitive aging phenomena which is misleading in a study where the oldest individual is 17years and pronounced effect of aging occurring only after that in the study species. So I suggest to frame the study of age as a predictor with experience effects instead. 2) Sex and rank effects cannot reliably be disentangled in a sample with only 2 low-ranking males. I suggest either restricting the investigation of rank effects to females or to omit sex as a predictor. I also would like to reiterate that a detailed description of methods applied for the construction of a dominance hierarchy along with a description of the data base (number of conflicts, spread across individuals and dyads) is required in a study that targets dominance effects. In addition to the issues raised by the reviewers I would also like to see the number of subjects participating in the study (21) to be mentioned more prominently instead of starting the paragraph with describing the composition of the group they live in (30). If you are able to address the critique in a satisfactory way and revise your manuscript accordingly, I will be happy to reconsider it for publication in RSOS.

- 1) As suggested, since our oldest individual was only 17 years old, we have clarified and reframed the focus of age as a proxy for experience (instead of the effect of cognitive aging). We have amended the introduction (l.70 to l.87 and l.174 of the manuscript with revisions), the methods (l.255) and the discussion (l.724-735) to focus on age as a proxy for experience and discussed how this might impact inhibitory control skills .
- 2) We restricted the investigation of rank effect to the females (l.258). The effect of rank was not considered in the models in which males and females were pooled together. An

additional analysis, for each task, was conducted for the effect of rank in models for which only females were considered (l.569, l.674, l.685, l.697 and l.708).

3) We added a description of the methods used to calculate David's scores (l.194) by the staff of the Medical Research Council Centre for Macaques (MRC-CFM), Porton Down, UK and the data obtained (see supplementary materials S2 David's Score sheet for the final data and regression analysis). The observations were based on agonistic interactions between individuals recorded ad libitum. Agonistic behaviours included threats (e.g. open mouth threat), displacements (i.e. a macaque approaches another who departs immediately), chases, and physical conflict (e.g. bite, slaps).

In addition a blind observer coded the video taken during the training and experiment and looked for the same agonistic interactions between the subject tested and other members of the group (see supplementary materials S2 Observation sheet for the results). The results of the observations confirmed the rank given by the staff of the facility.

4) The number of subjects participating in the study (21) was moved to the top of the paragraph (l.182):

“Subjects

All **21 subjects** (12 males, 9 females; aged from 3 to 17 years old, M = 8.1, SD = 4.05) participating in this study were adult rhesus macaques (*Macaca mulatta*) from 14 groups housed in the breeding colony of the Medical Research Council Centre for Macaques (MRC-CFM) in Porton Down, United Kingdom.”

Reviewer comments to Author:

Reviewer: 1

Comments to the Author(s)

The MS RSOS202102 reports a set of experiments aimed to investigate individual differences in inhibition in macaques. I found this study very interesting, with the important novelty of trying to assess different aspects of a cognitive function and different predictors of individual differences simultaneously. While this approach is common in psychology research in humans, it is quite rare in other species. For this reason, I feel that this paper would be an important contribution for research in the field. Below I suggests changes that should improve the presentation of the study.

Detailed comments

Abstract

L28 It is not clear the meaning of “the demographic factors did not influence the main components of inhibitory control”. Please provide more details or re-phrase.

We clarified the sentence by adding the following information (see abstract): “Hence, the

factors **age and sex were not associated consistently** with the main components of inhibitory control (...)"

Introduction

The introduction was clear and easy to follow. However, often it failed to acknowledge prior research in other animal species. I am confident that this issue can be fixed by discussing opportune references. I have suggested some studies performed in my lab and from other colleagues that might be of interest for this point.

We added further details and references of papers which studied individual differences in fish, birds, dogs and primates:

I.38: Brucks D, Range F, Marshall-Pescini S. (2017) Dogs' reaction to inequity is affected by inhibitory control. Scientific reports. **7**, 15802.

I.34: Lucon-Xiccato T, Bisazza A, Bertolucci C. (2020) Guppies show sex and individual differences in the ability to inhibit behaviour. Animal cognition. **23**(3), 535-43.

I.34: Lucon-Xiccato T, Montalbano G, Dadda M, Bertolucci C. (2020) Lateralization correlates with individual differences in inhibitory control in zebrafish. Biology letters. **16**(8):20200296

I.55: Lucon-Xiccato, T., & Bisazza, A. (2014) Discrimination reversal learning reveals greater female behavioural flexibility in guppies. Biology Letters, 10(6), 20140206.

I.37: Meier C, Pant SR, van Horik JO, Laker PR, Langley EJ, Whiteside MA, Verbruggen F, Madden JR. (2017) A novel continuous inhibitory-control task: variation in individual performance by young pheasants (*Phasianus colchicus*). Anim. Cogn. **20**, 1035-1047.

I.34: Montalbano, G., Bertolucci, C., & Lucon-Xiccato, T. (2020) Measures of inhibitory control correlate between different tasks but do not predict problem-solving success in a fish, *Poecilia reticulata*. Intelligence, **82**, 101486.

I.40: Völter CJ, Tinklenberg B, Call J, Seed AM. (2018) Comparative psychometrics: Establishing what differs is central to understanding what evolves. Philos. Trans. R. Soc. B **373**.

L11-15 I believe that this is a novel and important approach for animal behaviour studies. If possible, it would be beneficial to underlie it in the abstract.

At the end of the abstract, we added the information and changed the sentences to: "This study **adopts a novel approach for animal behaviour studies and gives new insight into** the individual variability of inhibitory control which is crucial to understand its evolutionary underpinnings."

L32-34 This sentence is a bit misleading because in the same species there is evidence of individual differences for several other IC tasks:

We added information about another task (the cylinder task) used in guppies (I.31): “For instance, in guppies (*Poecilia reticulata*), researchers, reported individual differences in two measures of inhibition of action. In these tasks, subjects needed to inhibit reaching directly for a prey through a transparent glass tube [30,31,32] or through a transparent cylinder [32].”

L38 There is an example in fish and one example in birds. I would recommend to add citations to previous studies on individual differences in mammals, including primates, as this study is pertinent to a primate species.

We added references of studies in dogs (I.37):

“Dogs also demonstrated individual differences in common inhibitory control tasks (detour and reversal learning tasks) [33].

33. Brucks, D., Marshall-Pescini, S., Wallis, L. J., Huber, L., & Range, F. (2017). Measures of dogs' inhibitory control abilities do not correlate across tasks. *Frontiers in Psychology*, 8, 849.

We have also made clearer that the study of Völter et al. reanalysed from an individual perspective comparative studies in several species including primates (I.40):

“Similarly, in a meta-analysis, Völter et al. [27], re-analysed, from an individual difference perspective, two large comparative studies [14,28]. **These studies measured, in several species, the inhibition of action (a detour-reaching task) and inhibition of a cognitive set (a A-not-B task, simplified variant of the reversal learning task in which the subjects are required to inhibit a previously rewarded behaviour to learn a new reward-contingency [14,28]). From the first study [14], they extracted and reanalysed performances of 15 species, and from the second study they reanalysed performances of 6 primate species [28].**”

14. MacLean EL et al. 2014 The evolution of self-control. *Proc. Natl. Acad. Sci. USA*, 111. E2140–E2148. (doi:10.1073/pnas.1323533111)

27. Völter CJ, Tinklenberg B, Call J, Seed AM. 2018 Comparative psychometrics: Establishing what differs is central to understanding what evolves. *Philos. Trans. R. Soc. B* **373**. (doi:10.1098/rstb.2017.0283)

28. Amici F, Call J, Watzek J, Brosnan S, Aureli, F. 2018 Social inhibition and behavioural flexibility when the context changes: A comparison across six primate species. *Sci. Rep.* **8**, 1–9. (doi:10.1038/s41598-018-21496-6)

L60 Perhaps it would be useful to cite this work finding the same effect in birds

We added the study of Kabadayi, Jacobs and Osvath, 2017, showing the same effect in ravens (I.77):

“For instance, ravens demonstrated a gradual increase in performance in the cylinder task through age [45].”

45. Kabadayi, C., Jacobs, I., & Osvath, M. (2017). The development of motor self-regulation in ravens. *Frontiers in psychology*, **8**, 2100.

L74-75 This sentence is a bit misleading. I think that there are evidence of sex differences in IC in animals, but not relating to distraction inhibition. Please rephrase this sentence and add references such as this paper cited below:

We clarified the sentence and added papers on sex differences in guppies (one using the tube task and one with the reversal learning task) (l.32-35 & l.55).

l.95: “In animals, male guppies had worse performances than females in an inhibition of action task, the transparent tube task, in which males attempted to attack the prey inside a transparent tube twice as often as females [30].”

30. Lucon-Xiccato, T., Bisazza, A., & Bertolucci, C. (2020). Guppies show sex and individual differences in the ability to inhibit behaviour. *Animal Cognition*, 1-9.

l.55: “Female guppies were also better at reversing a pre learned rule [37].”

37. Lucon-Xiccato, T., & Bisazza, A. (2014). Discrimination reversal learning reveals greater female behavioural flexibility in guppies. *Biology Letters*, *10*(6), 20140206.].

Methods

L150 The sample size seems generally OK compared to previous studies. However, I am not sure about the power to detect age differences. As other readers may have my same concern, I suggest to try to address this point or make a short comment in discussion.

We agree that the sample does limit us for testing cognitive aging or age differences (see response to the editors). However, when assessing individual differences, age is an important factor (l.70 to 87), and we think that we cannot ignore it in our analysis. We have also now added a sentence about the small sample size in the conclusion:

l. 868: “Inconsistent effect of age and sex could also be due to our small sample size. However, studies looking specifically at individual differences in guppies have used comparable sample sizes from 22 to 28 individuals.”

L160-161 From my understanding, the subjects were naïve of cognitive tasks. I suggest to make clear this point as very often primates used in cognitive studies have a quite long experience with cognitive experiments, and this can affect individual differences in performance.

We have now added this information in the Subject session of the General methods (I.201):

“ Eighteen of the subjects had already participated in a behavioural study involving looking at pictures [79]. However, none of them had experience with cognitive or touchscreen experiments.”

Experiment 1

L247 It would be helpful to use the terms Experiment 1, Experiment 2 etc also in introduction.

We added the information at the end of the introduction for each task (I.160):

I.160: “we conducted a Distraction task (**experiment 1**). (...), we conducted a Go/No-go task (**experiment 2**). (...) we conducted a reversal learning task (**experiment 3**).”

L271 Is there evidence that this species perceives the pictures as real conspecifics? If so, it would be nice to have a reference to strength the validity of the paradigm.

We added this information at the end of the Stimuli part for the first experiment (I.333). We used the model proposed by Fagot et al., 2014 [95]. According to this model, at first, monkeys are confusing the picture and the reality, but after repeated exposure, the animal could then consider the picture as equivalent or independent of the represented object.

“Several studies have demonstrated that rhesus macaques perceive, at first, pictorial stimuli of macaques, at first, as real conspecifics. For example, naive rhesus monkeys reacted to pictures of conspecifics with retreat, threat responses and vocalisations [93,94, for review see 95].”

95. Fagot J, Thompson RK, Parron C. 2010 How to read a picture: Lessons from nonhuman primates. Proc. Natl. Acad. Sci. 107, 519–20. (doi:10.1073/pnas.0913577107)

L313 This is the type of previous experimental experience that I asked to address above.

This point was addressed above by adding the information at the end of the Subjects section (I.201)

L319 Was there an improvement (learning) over trials and sessions?

The GLMM shows that there were better control scores as trials and sessions increased, so we have now added this information in the results section (I.399):

“Macaques demonstrated a better control scores as trial and sessions increased (see Table 1). “

L336 Please give more details about these effects. Are they a form of learning to solve the task more efficiently with practice?

Macaques demonstrated better control scores as trials and sessions increased through

potential habituation to the pictures. We have now added a paragraph in the conclusion (l.443):

“Macaques demonstrated better control scores as trials and sessions increased through habituation. Similar to the current study, in [92,93], rhesus macaques first reacted with vocalisations, threats and retreats to the first presentation of slides of conspecifics’ faces. However this behaviour did not persist, as macaques realised that the stimuli were only pictures. Fagot et al. [95] described in his model, a first phase of confusion between the picture and reality, but after repeated exposure, the animal could then consider the picture as equivalent to the represented object or process the picture and the object as independent.”

There are several small spacing errors in the result section of this experiment (eg. l.346).
The spacing errors have been resolved.

L374 The example of guppies is apparently a form of inhibition of action. If correct, this may be explicitly discussed (maybe also in the discussion of exp. 2).

Discussion l. 409 evolution etc. and for exp 2 l. 535

We explicitly discussed the example of Guppies in inhibition of action tasks in the discussion of exp.1 (l.453):

“In the literature, better inhibitory control skills in females were also found in the Go/No-go task [51] and in other tasks of inhibition of an action such as the cylinder task [30] and the tube task [30]. But also in a task of cognition inhibition, the reversal learning task [35,101].”

We also discussed this sex difference in discussion of experiment 2 (l.599):

“Surprisingly a study in guppies also demonstrated sex differences in tasks of inhibition of action, the cylinder and the tube task [30], with a sample of only 14 females and 14 males. Authors suggested that fish might not possess the same cognitive process or neural circuits underlying inhibition than mammals. This sex difference in guppies might be due to a specific mating system, the males having been selected for reduced behavioural inhibition.”

Experiment 3

I agree that the reversal learning task requires some form of inhibition. However, according to many authors it also requires flexibility, which is often considered an executive function diverse from inhibitory control (Diamond, 2003). I suggest to acknowledge the possible contribution of flexibility in the results of reversal learning task.

We have amended the discussion of experiment 3 and added (l.733):

“This age difference could be explained by the possible contribution of flexibility (also part of executive functions) diverse from inhibitory control [19].”

General discussion

Some of the subjects performed more than one experiment. This theoretically allows the authors to test for correlations between task performances, which provides a measure of inter task individual differences. Is it possible to add this analysis and use it for the general discussion?

In another submitted paper we looked at the repeatability of performances between tasks. So we added this information in the discussion (l.861):

“In another study (Loyant et al., submitted) we also looked at the repeatability of performances between tasks. We found consistency of performance between the inhibition of a distraction and the inhibition of an action this was not found for inhibition of a pre-learned rule which strengthens the multifaceted structure of inhibitory control.”

L654 The study by Montalbano et al. (reported above) showing positive covariation between two different measures of inhibitory control can be useful here.

We have now added this study in the discussion (l.807):

“Montalbano et al. [32], demonstrated individual differences in guppies and showed positive covariation between two different measures of inhibition of action (the tube task and the cylinder task).”

L664 It has been recently proposed that similar individual differences might be related to personality. I am wondering if this should be addressed here.

We amended the discussion and added another study (l.844):

“Besides, observed variation may reflect individual differences (e.g. motivation or personality) or idiosyncratic task requirements (e.g. context of the task or salience of the reward) with only a small proportion of the variation capturing changes in the studied process [23,39,120]. **For example guppies having bolder personalities showed greater inhibitory control abilities in the tube task [86].**”

86. Lucon-Xiccato T, Montalbano G, Bertolucci C. 2019 Personality traits covary with individual differences in inhibitory abilities in 2 species of fish. *Curr. Zool.* **66**, 187–195. (doi:10.1093/cz/zoz039)

Reviewer: 2

Comments to the Author(s)

This manuscript examines three components of executive function in 21 rhesus macaques using cognitive tasks implemented with touch screen computers. The main goals are to examine different sources of individual variation as well as examined the structure of cognitive performance. The paper makes a good case for the importance of studying

individual variation, something that has been a major focus of work in human cognitive sciences but less often a focus of comparative studies. While documenting individual variation is valuable, most of the time the real puzzle is explaining it. Unfortunately, I am not convinced the current paper appropriately tackles that second component, and I am also not sure the methodological approach is ideal for capturing (repeatable) individual variation as described. In addition, I have several clarification questions regarding what the specific procedures and analyses were.

1. Aims, hypotheses and predictions

The introduction mentions multiple potential sources of variation (pg. 2-4), but it seems like the devil is in the details concerning which kinds of self-control tasks are being discussed. In particular, the literature cited here deals with a fairly wide range of decision-making and executive function tasks, some of which seem quite related to the current tasks (e.g., age effects in reversal learning) but some of which are not as clearly linked (e.g., sex differences in sensation seeking or risk-taking), and which further vary by species. This makes the subsequent hypotheses for sources of individual variation in the specific tasks appear not appear well-explained.

We have now amended the introduction to review the effect of each factor on tasks related to only common inhibitory control tasks. The following sentences, which can lead to confusions, have been removed (l.88):

“Human studies suggest that men may be less able to control inappropriate behaviours than women [50–53]. For instance, men were reported to be more sensation seeking and more frequently engaging in risk-taking behaviours than women [53–54]. In a study with children, girls were better in behavioural self-control than boys at a very young age [55].”

For example, why is a sex difference (but not an age or rank difference) predicted in the Stroop task? This hypothesis shapes the analytic choices, as age and rank are not even examined in this study. In contrast, sex, age, and rank are predicted to impact performance on the go/no-go task (based on previous studies with various species, not a-priori theoretical hypotheses), leading to a different set of models even though the same subjects (mostly) are tested. Finally, only age is examined in the reversal learning task following the same kinds of logic. Overall, this makes the analysis procedures for each task appear somewhat arbitrary.

We have now made it clearer in the introduction that for each task all factors are tested (sex, age and rank, l.136-l.146), we have also amended the hypothesis that each factor should have an effect on each task (l.168-l.179):

“We hypothesised that the different factors would all have an effect on the 3 main components of inhibitory control.”

Finally, one of the major stated goals of the paper is to examine the structure of inhibitory

control. First off, I would point out that typically in psychology such a question would be addressed with a factor analysis (e.g., see Miyake et al 2000 Cognitive Psychology for an examine from human executive functions, or MacLean et al 2017 Animal Behaviour for a similar approach from animal cognition). This allows looks at common versus shared variance across different measures to infer the latent structure of these cognitive processes. However, this study likely has too few individuals (and too few independent task measures) to do this properly. Instead, the paper seems to take the approach of seeing if the same age-, sex-, and rank- effects account for performance across the different contexts. However, currently the manuscript does not report clearly parallel analyses testing for all of these effects in each task (as described above) which means this is not clearly assessed.

We have now made it clear that we have tested each factor for each task (see above).

More generally, I think there is a real theoretical question of whether looking at shared age-, sex-, or rank- effects across tasks is actually good evidence relevant to understanding the structure of cognitive abilities in the first place. While I think the question is interesting, I would say this is the weakest output of the paper and I would urge either taking a more well-understood approach from cognitive science to answer this question or else providing more clear justification for why this approach is valid.

We amended the core of the title, the introduction as well as the hypotheses to clarify that we looked at the effect of each factor on the 3 main components of inhibitory control but not directly into the structure of inhibitory control.

The previous title was : “Age and sex individual differences reveal a multifaceted structure of inhibitory control in non-human primates”. We amended it to : **“Heterogeneity of performances in several inhibitory control tasks: male rhesus macaques are more easily distracted than females”**.

We hypothesised that there will be an effect of each factor on each task performance (l.147-151):

“Therefore, the aim of this study was three-fold: (1) to systematically demonstrate individual variability in the three main components of inhibitory control in non-human primates (2) to investigate the most common causes (age, sex and rank) of these individual variation (3) **to determine if these influencing factors have coherent effects on the 3 main components of inhibitory control.**

l.168 “We hypothesised that the different factors would all have an effect on the 3 main components of inhibitory control.”

2. Study population and sample size

While I appreciate the difficulty in acquiring a larger sample size of animals, 21 individuals is a small sample size for testing questions about individual differences. Indeed, this kind of sample size limitation is exactly one reason why there are so few studies of individual

variation in nonhuman cognition. Nonetheless, I think it would be important to better justify how this sample size is appropriate for the questions set up here. For example, are there past studies showing age, sex, or rank effects in rhesus monkeys with comparable sample sizes? Could the authors perform a power analysis?

We performed a power analysis using the mean performances obtained in our study to compare the Distraction control score of males and females. For instance, to reach a Power analysis of 0.8 for the first task, the sample size should be of 115 macaques. Therefore we understand that our limited sample size considerably reduced the strength of our analysis. Unfortunately in non-human primate studies, the sample size is often limited by the number of individuals available; this holds true for both captive and wild populations. For instance, from a meta-analysis of 574 primate cognition papers, Many Primates *et al.*, 2019, found a median sample size of 7 individuals per study. However, we believe that cognitive studies on non-human primates, even with a limited number of subjects, are still crucial to gather precious information on the evolution of cognition. In our study, we used as many trials as possible on each of our subjects and treated individuals as a “replication unit”, which should increase the chance to find a statistical effect as recommended in Farrar, Boeckle and Clayton (2020).

ManyPrimates, Altschul, D., Beran, M.J., Bohn, M., Caspar, K.R., Fichtel, C., Försterling, M., Grebe, N.M., Aguilar, A., Kwok, S.C., Llorente, M., Motes-Rodrigo, A., Proctor, D., Sánchez-Amaro, A., Simpson, E., Szabelska, A., et al., (2019). Collaborative open science as a way to reproducibility and new insights in primate cognition research. *Japanese Psychological Review* **62**, 205-220.

Farrar, B. G., Boeckle, M., & Clayton, N. S. (2020). Replications in Comparative Cognition: What Should We Expect and How Can We Improve? *Animal behavior and cognition*, 7(1), 1–22. <https://doi.org/10.26451/abc.07.01.02.2020>

More particularly, here are some examples of sample size in studies looking at individual differences in animals. The sample size in previous work is comparable to the sample size in this study.

For instance papers looking at individual differences in guppies (cited in the text):

- 22 adult females were tested in the following paper:

Montalbano, G., Bertolucci, C., & Lucon-Xiccato, T. (2020). Measures of inhibitory control correlate between different tasks but do not predict problem-solving success in a fish, *Poecilia reticulata*. *Intelligence*, 82, 101486

- 28 guppies in this papers

Lucon-Xiccato T, Montalbano G, Dadda M, Bertolucci C. Lateralization correlates with individual differences in inhibitory control in zebrafish. *Biology letters*. 2020 Aug

26;16(8):20200296.

Papers looking at sex difference:

- 28 guppies for the following papers (cited in the text):

Lucon-Xiccato T, Bisazza A, Bertolucci C. Guppies show sex and individual differences in the ability to inhibit behaviour. *Animal cognition*. 2020 May;23(3):535-43.

Lucon-Xiccato, T., & Bisazza, A. (2014). Discrimination reversal learning reveals greater female behavioural flexibility in guppies. *Biology Letters*, 10(6), 20140206.

Miletto Petrazzini ME, Bisazza A, Agrillo C, Lucon-Xiccato T. Sex differences in discrimination reversal learning in the guppy. *Animal cognition*. 2017 Nov;20(6):1081-91.

Papers looking at age difference:

- 5 raven chicks were tested:

Kabadayi, C., Jacobs, I., & Osvath, M. (2017). The development of motor self-regulation in ravens. *Frontiers in psychology*, 8, 2100.

- When looking at age difference in rhesus in other cognitive process, 12 monkeys were used in the following study:

Moss MB, Rosene DL, Peters A. Effects of aging on visual recognition memory in the rhesus monkey. *Neurobiology of aging*. 1988 Jan 1;9:495-502.

- 13 monkeys used:

Moore TL, Killiany RJ, Herndon JG, Rosene DL, Moss MB. Impairment in abstraction and set shifting in aged rhesus monkeys. *Neurobiology of aging*. 2003 Jan 1;24(1):125-34.

- 26 for this study in age and sex difference in rhesus macaques:

Lacreuse A, Herndon JG, Killiany RJ, Rosene DL, Moss MB. Spatial cognition in rhesus monkeys: Male superiority declines with age. *Hormones and Behavior*. 1999 Aug 1;36(1):70-6.

- For studies using touch screen in rhesus, 10 males were tested

Bethell EJ, Holmes A, MacLarnon A, Semple S. Emotion evaluation and response slowing in a non-human primate: new directions for cognitive bias measures of animal emotion? *Behavioral Sciences*. 2016 Mar;6(1):2.

- 8 males were used in the following study on rhesus macaques:

Bethell, E.J., Holmes, A.E., MacLarnon, A., & Semple, S. (2012). Evidence That Emotion Mediates Social Attention in Rhesus Macaques. *PLoS ONE*, 7.

To the best of our knowledge there are no articles looking specifically at the effect of individual differences and rank in rhesus monkeys.

We discuss this sample size limitation in the discussion (l.868):

“However, studies looking specifically at individual differences in guppies had a sample size going from 22 to 28 individuals [30–32,37,100]. A low sample size, a common limitation when working with primates, might have decreased the power of our analysis [111,67].”

I also have some questions about the monkeys. it seems like the final sample of 21 was drawn from multiple groups—are these of species typical size; do they include mixed sex or age individuals? Who do the subject monkeys live with? This kind of information about their social experiences is important for interpreting the findings about sex, age, and rank differences. In addition, little information about how the dominance rank information was acquired is provided. Was this just the opinion of staff, or stemming from systematic behavioural observations? As currently written it sounds like it may have been staff opinion. If so, how reliable is this metric? Since 15 of 21 final sample of subjects are reported to be high-ranking, this seems rather skewed.

We added the information about the group in the Subjects section (l.185-188):

“The subjects were taken from 12 mixed group from 9 to 20 individuals with a mean of 15 individuals, they were constituted of 1 male and several females and younglings. One group was constituted of only males and one group of only female.”

As stated in response to the editor above, rank calculations were based on David’s scores and this information has now been added for clarity (l.194).

3. Methods

One general comment is that the validation of the tasks used here appears to be unpublished (e.g., it is cited as Loyant et al., submitted), and I have several comments on whether these tasks are actually assessing what they aim to.

Stroop task: For example, it’s not clear to me that Study 1 (Stroop task) is properly considered a Stroop task. The task used with monkeys involved touching a target red rectangle, and in different conditions various distractors (a neutral object, a neutral conspecific, or a threatening conspecific) also appear on the screen. The analyses examined whether these distractors influenced reaction time. Yet the human Stroop task is not focused

on whether people are distracted by irrelevant extra stimuli, but on how a stimulus feature can distract processing another stimulus feature. For example, imagine the task is to report how many words appear on a screen, and then seeing a screen with a word printed three times. While it might be easy to correctly report “3” if the word printed three times is “cat,” it is more challenging to report “3” when the actual word printed three times is ‘four’. This conflict between different dimensions of the stimuli is key to the Stroop task, and the monkey task does not seem to have this element of needing to shift attention between different aspects of the stimuli.

We called it Modified Stroop task in the sense that the subject needed to inhibit an interference from a stimulus to perform a task. To avoid potential confusion, we have now changed it to Distraction task. The subject had to inhibit their emotional reaction to the distractor to perform the task.

Go/no go task: In the go/no-go task, it appears that monkeys completed 10 sessions of 20 trials each (pg. 19) but only the last 3 were analysed.

Apologies, there was a mistake in the text, the 5 sessions were analysed in the models. The error has been corrected in the analysis section (l.555).

It is unclear if these took place on the same day or not. One important consideration is whether the monkeys had actually learned the difference between the go and no-go options given that it appears they had to acquire this only through direct experience with choosing the options. Is there some other validation that the animals understood the task?

There were large individual differences between macaques in this task. For the last session, when considering a success as not touching the No-go stimulus, 3 macaques were above 80% of success (i.e. not touching the No-go stimulus significantly above chance) and 5 below 50% of success (l.582). So, yes, we can validate that macaques are able to understand the task and therefore are able to differentiate both stimuli. This task is particularly difficult as its aim is to create a dominant automatic answer (to touch the screen with a Go stimulus presented 75% of time), and the wrong answer (to touch the No-go stimulus, which is automated and hard to control). Even in humans this task leads large individual differences between subjects and study, even if the task has been explained and is understood from the beginning. For instance:

- In 59 adult the mean percent of correct answers for the No-go trial was 60.3% (Hirose S, Chikazoe J, Watanabe T, Jimura K, Kunimatsu A, Abe O, Ohtomo K, Miyashita Y, Konishi S. Efficiency of go/no-go task performance implemented in the left hemisphere. *Journal of Neuroscience*. 2012 Jun 27;32(26):9059-65)

- In 69 adults: 75% to 92.6% of success in the No-go trial (Benikos N, Johnstone SJ, Roodenrys SJ. Varying task difficulty in the Go/Nogo task: the effects of inhibitory control, arousal, and perceived effort on ERP components. *International Journal of Psychophysiology*. 2013 Mar 1;87(3):262-72.)

- In 54 adult the accuracy was 85% in No-go trials (Littman R, Takács Á. Do all inhibitions act alike? A study of go/no-go and stop-signal paradigms. PLoS One. 2017 Oct 24;12(10):e0186774.)

- in 120, 3 years old children, there was a 70% accuracy in No-go trials (Simpson A, Riggs KJ. Conditions under which children experience inhibitory difficulty with a “button-press” go/no-go task. Journal of Experimental Child Psychology. 2006 May 1;94(1):18-26)

Finally, the specifics on trial, sessions, and testing days for the different tasks is a bit confusing and spread out over multiple sections. It would be helpful to consolidate this so the reader can have a clearer picture of what actually happened. For example, the general method indicates that the tasks were conducted in the same order for each subject and that once all the sessions of a task were complete, the next task was conducted on the next testing day (page 8-9) which would imply that monkeys did each task on separate days and completed multiple “sessions” per task on the same day. Then, the Stroop task seems to have involved 3 sessions of 36 trials each, with each session split into 6 blocks (page 12). What is the significance of sessions versus blocks? Did the monkey get a pause? This matters both for understanding the actual procedure, as well as for interpreting the repeatability scores reported for the tasks as discussed below

We have now amended and clarified the procedure section (l.235):

“There was a 5 to 10 minutes break between each sessions to allow the subjects to refocus on the task. (...). If the subject stayed inactive for more than 5 min or lost its attention the experiment was stopped and the remaining sessions were finished the next testing day, if the subject did not participate for three testing days in a row the subject was excluded from the task.”

Blocks allowed to pool together pictures having a similar emotional valence to increase the effect of the stimulus (l.348). As there was a break without testing between each sessions it allowed the individuals to come back to an emotional baseline before starting a new session.

4. Analyses and results interpretation

I already discussed above how different analysis procedures are used on the different tasks such that the impact of age, sex, and rank are tested in the go/no-go task, but only sex is mentioned in the Stroop task and only age in the reversal task.

This point was confusing for the reader and it has been addressed above, all 3 factors were considered for each task.

Another issue concerns the repeatability analyses. For example, it is reported that individual macaques exhibited significant repeatability in these tasks. But, how meaningful is this if all

trials/blocks/sessions occurred on the same day? Repeatability in behavioural metrics, for example, generally look at whether there is a core “personality” dimension that predicts behaviour across contexts and over time (for example, Tkaczynski et al 2020 Royal Society Open Science). It seems like a comparable claim for stable individual differences in cognitive performance would require testing the same individual at least on different days. Assuming that cognitive performance can be influenced by both stable individual differences but also situational factors (what’s going on in the social group, the monkey’s emotional state, hunger, etc) these situational factors will be common across the testing sessions from the same day and thus could account for apparent repeatability in analyses.

To check this potential issue we reran the analysis by taking into account only macaques tested on several days, which did not change the significance of the results (1.293).

“To maximize individual variations and validity of the repeatability analysis we only included subjects which performed the tasks on different days (N = 16 for the Distraction task, N = 15 for the Go/No-go task and N = 19 for the reversal learning task).”